# A myosin II nanomachine mimicking the striated muscle

Irene Pertici [1], Lorenzo Bongini[1], Luca Melli[1,4], Giulio Bianchi[1], Luca Salvi[1,5], Giulia Falorsi[1], Caterina Squarci[1], Tamás Bozó [2], Dan Cojoc[3], Miklós S.Z. Kellermayer [2], Vincenzo Lombardi [1] & Pasquale Bianco [1]

The contraction of striated muscle (skeletal and cardiac muscle) is generated by ATP-dependent interactions between the molecular motor myosin II and the actin filament. The myosin motors are mechanically coupled along the thick filament in a geometry not achievable by single-molecule experiments. Here we show that a synthetic one-dimensional nanomachine, comprising fewer than ten myosin II dimers purified from rabbit psoas, performs isometric and isotonic contractions at 2 mM ATP, delivering a maximum power of 5 aW. The results are explained with a kinetic model fitted to the performance of mammalian skeletal muscle, showing that the condition for the motor coordination that maximises the efficiency in striated muscle is a minimum of 32 myosin heads sharing a common mechanical ground. The nanomachine offers a powerful tool for investigating muscle contractile-protein physiology, pathology and pharmacology without the potentially disturbing effects of the cytoskeletal—and regulatory—protein environment.

---

[1] PhysioLab, University of Florence, Florence 50019, Italy. [2] Department of Biophysics and Radiation Biology, Semmelweis University, Budapest H-1094, Hungary. [3] IOM-CNR, Trieste 34149, Italy. [4] Present address: F. Hoffmann-La Roche Ltd, Basel 4053, Switzerland. [5] Present address: Department of Biochemistry, University of Munich, Munich 81377, Germany. Correspondence and requests for materials should be addressed to V.L. (email: vincenzo.lombardi@unifi.it)

Myosin II is a dimeric mechanoenzyme that uses the free-energy change associated with the binding and hydrolysis of ATP in either of its motor domains (the heads) to generate force and displacement along filamentous actin via a 10-nm working stroke[1]. In the striated-muscle sarcomere, the motors emerge as two antiparallel arrays from a bipolar thick filament. As a result, steady force and shortening in the sarcomere are generated by cyclic ATP-driven interactions of the heads that pull the actin filament from the opposite extremities of the sarcomere towards the centre[2–4]. The motors of each half thick filament are mechanically coupled via the thick filament backbone, which allows for steady contraction in spite of the fact that the individual myosin heads interact with actin only briefly and remain detached throughout most of their ATP hydrolysis cycle. The duty ratio, that is the fraction of time the myosin head is bound to actin during its kinetic cycle, is as low as 0.05 in an unloaded contraction, which implies that no more than 5% of the available heads are attached to actin at any moment[5,6]. The energetic cost of low-load contraction is thus minimised, due to the strain dependence of the chemo-mechanical steps of the ATPase cycle in the attached heads[2,7]. In the intact cell environment, the half-sarcomere is able to work across a wide range of externally applied loads by tuning the number of heads attached to actin in proportion to the global filament load, while the force of the individual attached head is maintained similar to that developed under isometric conditions by the progression of the working stroke[8]. Cell-mechanics experiments, however, cannot reveal the motor coupling mechanism because of the large ensemble of the motors and the unavoidable presence of mechanically coupled cytoskeletal proteins and filaments. By contrast, single-molecule mechanics experiments on purified proteins[9–11] provide insights into the motor function, but they suffer from the intrinsic limit in that they cannot detect the unique performance emerging from the collective nature of motor action within the architecture of the half-sarcomere. Only myosin II motors working in ensemble can fit the task, but so far the various attempts failed to produce steady force and shortening in solution with physiological [ATP][12–15].

Here we demonstrate the performance of a one-dimensional machine titrated to contain the minimum number of motor molecules needed to reproduce the collective mechanics of muscle myosin II in situ. The mechanical output of the ensemble of myosin II molecules purified from the psoas muscle of the rabbit and brought to interact with a correctly oriented actin filament is measured by means of counter-propagating dual laser optical tweezers (DLOT) in either position or force clamp. The design of the nanomachine ensures that, like in the three-dimensional lattice of the sarcomere, the array of motors interacting with the actin filament lies on a plane parallel to the actin filament, overcoming a main limit of all existing designs[12–15]. In 2 mM ATP, the nanomachine reproduces the steady force and shortening velocity typical of the isometric and isotonic contractions of the sarcomere in vivo. The power of the nanomachine is interpreted with a simple model, the mechanical–kinetic features of which are constrained by preliminarily fitting the performance of fast mammalian muscle. We find that, to attain the efficiency of striated muscle with a one-dimensional machine, a minimum of 32 myosin heads sharing a common mechanical ground must be available for interaction with the actin filament.

## Results
### Constructing a one-dimensional myosin II nanomachine.
The nanomachine is an ensemble of fast vertebrate skeletal-muscle heavy meromyosin (HMM) molecules attached onto a single-mode optical fibre chemically etched to a diameter of ~4 μm. A three-way piezoelectric nanopositioner brings the HMM ensemble to interact with a fluorescently labelled actin filament (length range 7–11 μm), attached with the correct polarity (bead-tailed actin (BTA))[16] to a polystyrene bead held with DLOT[17] in the middle of a 180-μm-deep flow chamber (Figs. 1a and 2a and Supplementary Fig. 1). Thus the single actin filament selects a functional one-dimensional array out of the fibre-bound HMM molecules. The mechanical output of the nanomachine is measured by means of the DLOT, which acts as a force transducer (range 0.5–200 pN, compliance 3.7 nm pN$^{-1}$) and the nanopositioner carrying the HMM-coated fibre, which acts as a displacement transducer (range 1–75,000 nm). The system operates either in position- or in force-feedback mode[18] (see Methods and Supplementary Fig. 2). Methylcellulose (0.5% w/v, 400 cP) was present in the solutions used for the experiment to inhibit the lateral diffusion of F-actin[6], thereby minimising the probability that, in 2 mM ATP, the interaction terminates during mechanical protocols that minimise the number of actin-attached heads. All experiments were conducted at room temperature (23 °C).

### Number of myosin molecules available for actin interaction.
The number of HMM molecules on the fibre surface able to interact with the actin filament is initially determined by measuring the number of mechanical rupture events in ATP-free solution[19,20] (Fig. 1). First the actin filament is brought to interact with the HMM-coated surface to form rigor bonds, then it is pulled away at constant velocity in a direction orthogonal to the motor surface (z, parallel to the axis of the trapping laser, velocity 25 nm s$^{-1}$) to break the rigor bonds one at a time while avoiding that the detached HMM binds back (Fig. 1a). The number of rupture events in a single mechanical trial increases with the [HMM] used to coat the fibre (Fig. 1b) according to a relation that attains a saturating value of 8.2 ± 1.2 at [HMM] = 100 μg ml$^{-1}$ (Fig. 1c), which is the concentration adopted for the experiments. The length of the functional nanoarray, estimated by summing the distances travelled following the rupture events (Supplementary Fig. 3) is 523 ± 175 nm (mean ± SD), one order of magnitude smaller than the average length of the actin filament. This architecture of the machine implies that for a given [HMM] the measured number of rupture events does not significantly change from experiment to experiment, therefore there is no need to normalise the mechanical data by actin filament length.

### Isometric and isotonic performance of the nanomachine.
In experiments on the active nanoarray (Fig. 2), a solution with 2 mM ATP is flowed into the chamber. The BTA is brought towards the motor array by moving the nanopositioner in the z direction in position feedback. Once an acto-myosin interaction is established, the position is clamped and the ensemble of motors starts to develop force (F) (phase 1) up to a maximum steady value ($F_0$) (isometric condition). Subsequently, the control is switched to force clamp (phase 2) and a staircase of stepwise reductions in force (4 pN per step in the experiment in Fig. 2b and 5 pN in the experiment in Fig. 2c) separated by 1 s intervals is imposed (phases 3–6, isotonic conditions). The nanomachine responds with actin filament sliding in the direction of shortening at a constant velocity (V), which is greater the smaller the level of constant force between steps.

In Fig. 2b, the interaction is terminated after ~2 μm of total sliding because of force-feedback failure due to the interference of the optical fibre with the laser beam forming the optical trap. In Fig. 2c, following the fifth step that drops the force to almost zero (phase 7), the control is switched to position feedback and the

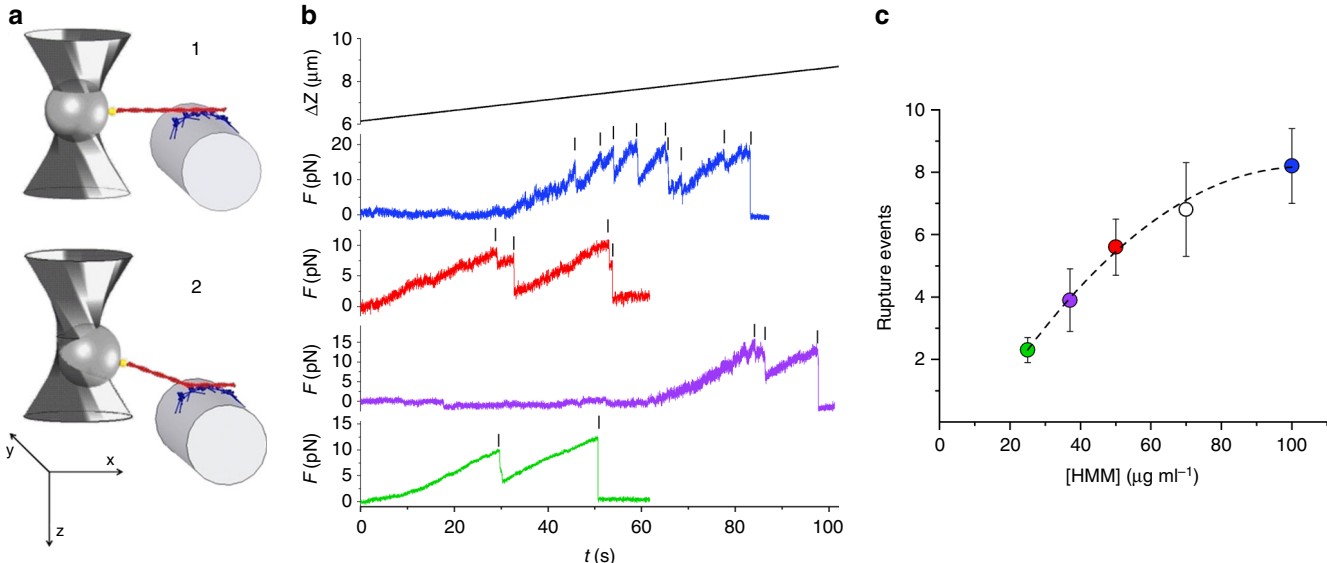

**Fig. 1** Assessing the number of HMMs available for actin interaction from rigor rupture events. **a** Schematics of the protocol: (1) formation of rigor bonds between the HMM array (blue) by the optical fibre (grey) and the actin filament (red) attached to the trapped bead (dark grey) via gelsolin (yellow); (2) optical fibre moved away from the actin filament in the direction ($z$) orthogonal to the actin–myosin interface. **b** Records of the nanopositioner movement (black, $\Delta Z$ velocity 25 nm s$^{-1}$) and force with different [HMM] (colours refer to different concentrations as indicated in **c**; green: 25 µg ml$^{-1}$; violet: 38 µg ml$^{-1}$; red: 50 µg ml$^{-1}$; blue: 100 µg ml$^{-1}$). The small vertical bars indicate the rupture events (force drop <2.5 pN, complete in <50 ms and not rapidly reversed), the last of which corresponds to complete detachment of the actin filament. **c** Relation between the number of ruptures per interaction and [HMM] (mean ± SD, $n = 14$)

force redevelops to the original $F_0$ value (phase 8), even if the total active sliding from the beginning of the interaction has attained 2.7 µm. A series of stretches and releases of ~500 nm superimposed on $F_0$ (phases 9–11) produces a transient ~0.7 $F_0$ increase (upon stretch) and reduction (upon release) of force followed by a recovery towards $F_0$ that demonstrates how remarkably constant is the active isometric capability of the nanomachine, regardless of the direction of actin filament sliding. In the absence of force pauses and fluctuations, both the initial rise of force and the force redevelopment following a large release (Fig. 3a, b, respectively) can be fitted with a single exponential (dashed lines) with $\tau = 0.15 \pm 0.09$ s (see also Supplementary Fig. 5).

$F_0$ from 46 experiments shows a Gaussian distribution with centre at 15.5 pN (Fig. 2d), as expected from the architecture of the machine that implies that for a given [HMM] the size of the array does not change from experiment to experiment and with the length of the actin filament. The relation between force and shortening velocity (F–V) is plotted in Fig. 2e, where points from individual experiments are grouped in classes of force 0.15 $F_0$ wide. Only three experiments contribute to the F–V point for $F < 0.2\ F_0$, as at these low forces the probability of actin filament detachment from the array increases up to a level at which the ability to maintain the interaction is lost even in the presence of 0.5% methylcellulose[6]. The dashed line is the fit of the F–V relation with Hill equation[21]: $(F + a)(V + b) = (F_0 + a)b$, where $a$ is the distance of the vertical asymptote from the $y$ axis and $b$ the distance of the horizontal asymptote from the $x$ axis. The ordinate intercept is the velocity of unloaded shortening ($V_0$) and is estimated by the fit as $3.40 \pm 0.45$ µm s$^{-1}$, which is 30% larger than the velocity of actin sliding ($V_f$) in in vitro motility assay (IVMA, Supplementary Fig. 4). $a/F_0$, the Hill equation parameter related to the curvature of the F–V relation[21] (see Methods), is $0.23 \pm 0.15$. The relation of power, $P$ ( $= F \cdot V$), versus $F$ (Fig. 2f) shows a maximum ($P_{max}$) of 5.4 aW at $F = 0.3\ F_0$.

**A kinetic model that explains nanomachine performance**. To understand how the performance of the nanomachine recapitulates muscle contraction at the molecular level, we use a simple mechano-kinetic model of the myosin motors made by three states (one detached, D, and two different force-generating attached states, A1 and A2, Fig. 4a). The stiffness of the attached head is 2 pN nm$^{-1}$[12,22,23]. As detailed in Methods, the rate functions for the state transitions and their strain-dependence (see Supplementary Fig. 6 and Supplementary Table 1) are first defined to simulate the mechanics and energetics of the fast skeletal muscle of mammals at the same temperature (23 °C)[24–27] and then used to fit the output of the nanomachine.

Myosin heads attach in state A1 and rapidly equilibrate with state A2 in which they perform the working stroke that accounts for the maintenance of steady force up to 5 nm shortening. This is in accordance with the force attained in phase 2 (T2) following step length perturbations in fibre experiments[28,29] (Supplementary Fig. 6b). Detachment from A2 is the rate limiting step in the cycle in isometric conditions, when the rate of ATP splitting is minimum (~11 s$^{-1}$ per myosin head at room temperature[10,24,27]) and the fraction of heads attached (the duty ratio) is maximum (~0.3)[23]. With the reduction of the load the rate of ATP splitting increases no more than 3 times[24], which cannot explain the corresponding increase in power[28]. Under these conditions, the maximum power can be predicted only by assuming that, during shortening, the attached myosin heads can rapidly regenerate the working stroke by slipping to the next actin farther from the centre of the sarcomere during the same ATPase cycle[10,28,30–34] (step "slip" in Fig. 4a). Detachment from either A1' or A2' (step 3') implies ATP hydrolysis (for more details, see Methods and Supplementary Fig. 6).

The simulation of the responses of the nanomachine requires taking into consideration that (i) the compliance in series with the motor array rises by more than two orders of magnitude from the value in the half-sarcomere (~10$^{-2}$ nm pN$^{-1}$[35]) to that of the

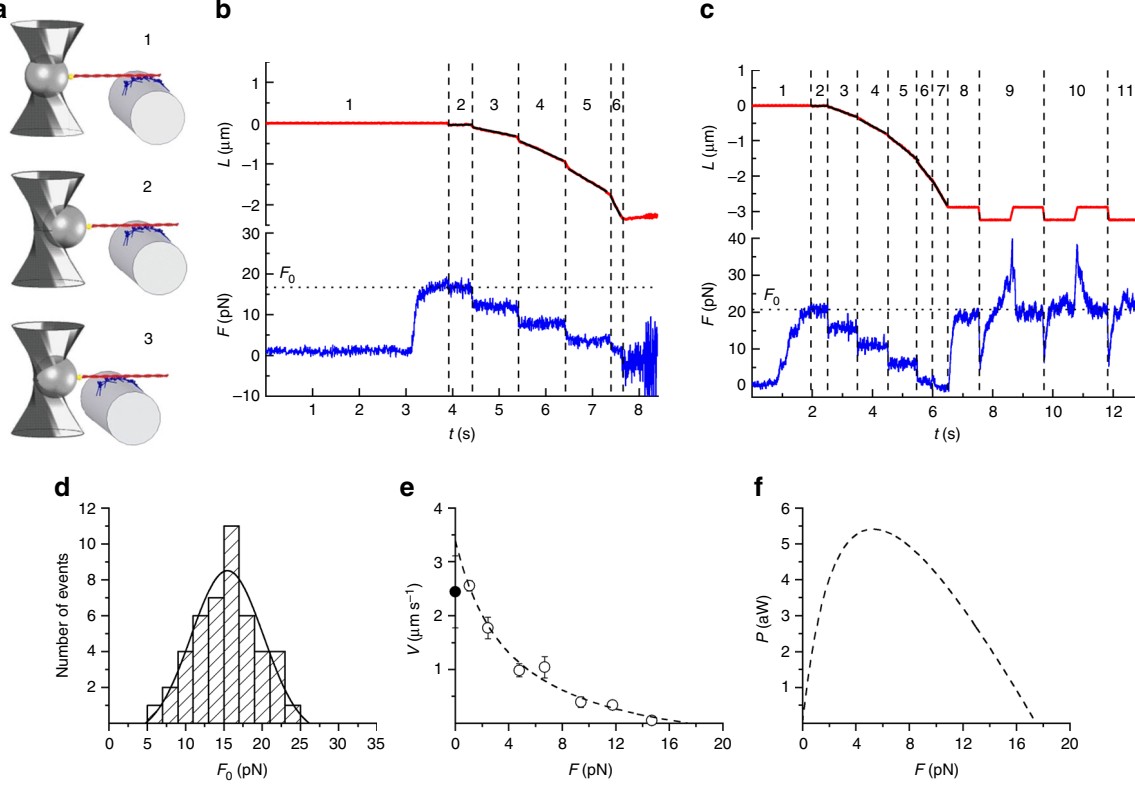

**Fig. 2** Mechanical output of the myosin-II nanomachine in 2 mM ATP. **a** Schematic representation of three snapshots during the phases of the interaction between the actin filament and the motors numbered as in **b**. Components of the system coloured as in Fig. 1a. **b** Recording of the relative sliding (red trace) and force (blue trace) during the interaction. Phase 1, following the formation of the first bonds between the actin filament and myosin motors, the force rises in position feedback to the maximum isometric value ($F_O$ ~ 17 pN). Phases 2–6, shortening response to a staircase of stepwise reductions in force (4 pN) separated by 1 s imposed in force feedback. **c** Another experiment in which in phase 1 in position feedback isometric force rises to $F_O$ (~21 pN) with some force fluctuations and in frames 2–7 the staircase in force feedback is made by 5-pN steps separated by 1 s (3–5) and 0.5 s (6, 7). In phase 8, after a total active sliding of 2.7 μm, the control is switched back to position feedback, and the force redevelops up to the original $F_O$. Phases 9–11, force response to a series of rapid lengthenings and shortenings of ~500 nm superimposed on $F_O$. In both **b**, **c**, black lines are superimposed on the length trace to better appreciate the slope. **d** Frequency distribution of $F_O$. Data are plotted in classes of 2 pN and fitted with a Gaussian (continuous line): centre = 15.5 ± 0.4 pN, $\sigma$ = 4.7 ± 0.4 pN (mean ± SEM, $n$ = 46). **e** Force–velocity ($F$–$V$) relation. The black filled symbol on the ordinate is the $V_f$ value from IVMA (see related Supplementary Fig. 4). The dashed line is Hill's hyperbolic fit to the data. **f** Power–force ($P$–$F$) relation from the fit in **e**

trap (3.7 nm pN$^{-1}$), and (ii) the motors are randomly oriented in the array, so that the force of HMM is reduced by a factor of two with respect to that on a correctly oriented myosin rod cofilament[10]. With the trap compliance in series, the addition or subtraction of the force contribution by a single head induces substantial sliding of the actin filament and a corresponding change in the strain of the other attached heads, undermining the condition of independent force generators that characterises the action of the myosin heads in isometric contraction in situ (Supplementary Fig. 7, upper row left panel). The strain-dependent kinetics of the attached heads is governed by the push–pull experienced by all the other attached heads when actin slides away towards the bead for the addition–subtraction of the force contribution by one head (upper row, right panel), and under these conditions, the isometric ATPase rate ($\varphi_0$) and slipping rate ($r_{s,0}$) increase (Table 1). During isotonic shortening, the perturbation of the attached head induced by the force contribution from each single head is still present (Supplementary Fig. 7, lower row), but the effect on both $\varphi$ and $r_s$ vanishes (Table 1), because the reduction of head strain increases anyway the head cycling kinetics. As for the correction for the effect of the random orientation of the heads in the array, the force of each attached head of the nanomachine is assumed to vary, as a

function of the angle formed with the correct orientation, by a factor ranging between 1 (0°) and 0.1 (180°)[10], which results in a reduction of $F_O$ generated by the randomly oriented attached heads by 45% and a corresponding leftward shift in the calculated $F$–$V$ relation for the half-sarcomere (Supplementary Fig. 6, dotted line). With these assumptions, the model is tested for the ability to simulate the machine performance by keeping unaltered all the mechano-kinetic parameters selected for muscle half-sarcomere and using as the only free parameter the number of heads available for the actin interaction, $N$. According to in situ mechanical and structural evidence that in active muscle the two heads of each dimer work independently[36], $N$ is twice the number of molecules and thus is 294 in the half-thick filament.

The $F$–$V$ and $P$–$F$ relations of the nanomachine are best fitted with $N$ = 16 (Fig. 4b, c: continuous line, model; symbols and dashed line, experiment). Notably, $N$ = 16 is expected from the number of HMM molecules (8) estimated from rupture events in rigor (Fig. 1), if, as in situ, also in the machine at 2 mM ATP and thus at a correspondingly low duty ratio (≤0.3), both heads of each dimer work independently. It must be noted that this conclusion appears in contrast with single molecule experiments showing a cooperative behaviour between the two heads of the myosin dimer at low ATP[37].

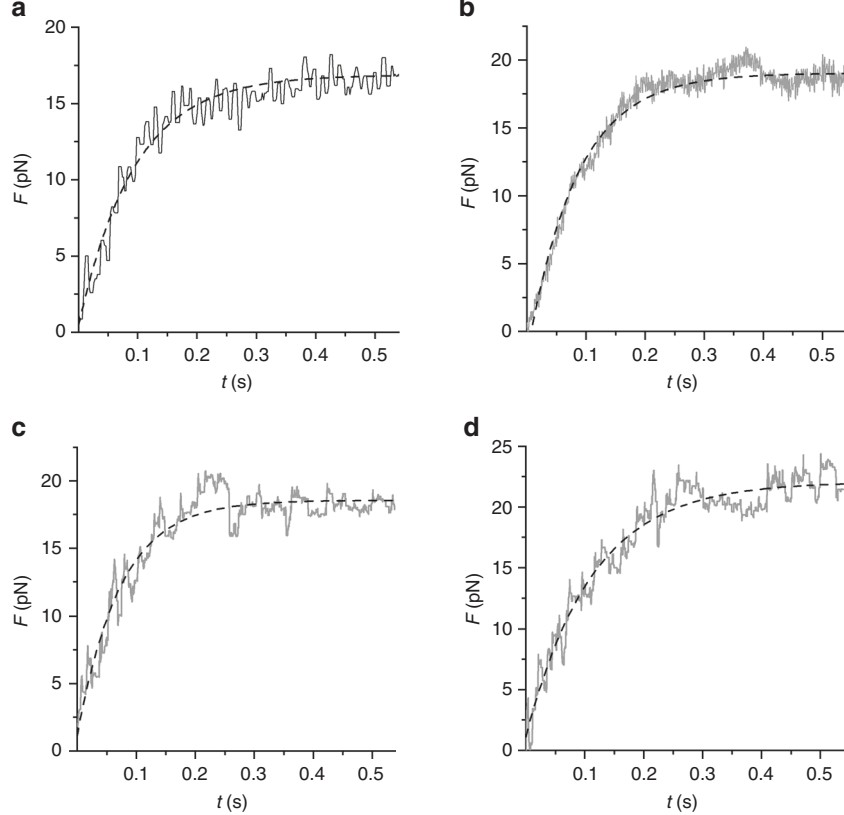

**Fig. 3** Rate of force development. Upper row: time course of initial force development (**a**, Frame 1 from Fig. 2b) and force redevelopment following a large release (**b**, Frame 8 from Fig. 2c). The different trace thickness in **a** and **b** is due to different acquisition rates (500 and 5000 points s$^{-1}$, respectively). Dashed lines are single exponential fits. Lower row: Model simulation of initial force development (**c**) and force redevelopment following a large release (**d**), using 16 available myosin heads and a series compliance of 3.7 nm pN$^{-1}$. Dashed lines are single exponential fits. $\tau$, calculated from fits to experimental traces as in **a** and **b** (0.15 ± 0.09 s, mean ± SEM, $n = 34$), is not significantly different from that calculated from fits to simulated traces in **c** and **d** (0.11 ± 0.06 s, mean ± SEM, $n = 34$) (see related Supplementary Fig. 5)

In Fig. 4d, the various simulated mechanical parameters are plotted as a function of $N$ (filled circles). The relations for $F_0$ and $P_{max}$ intersect the dashed lines (experimental values) at $N = 16$. By contrast, independent of $N$, the predicted $V_0$ remains greater than the experimental values. This discrepancy is likely due to a $V_0$-quenching effect caused by the random orientation of the motors. A similar conclusion has been drawn for actin sliding velocity in IVMA experiments[10,38,39], but that conclusion was contradicted by other evidence[5,6,40].

The time courses of force rise and redevelopment (upper row in Fig. 3) are reproduced with a remarkable agreement by the model simulation with the series compliance equal to 3.7 nm pN$^{-1}$ (lower row) (see also Supplementary Fig. 5). Noteworthy, as detailed in Supplementary Fig. 5, the number of interactions necessary to rise the force to the maximum isometric value against the trap compliance is compatible with the number allowed in probabilistic terms by an ensemble of 16 independent myosin heads, further proving the solidity of the kinetic and methodological analysis used for interpreting the nanomachine performance.

The number of attached heads ($N_a$) predicted by the model during isometric contraction of the nanomachine fluctuates between 5 and 6, each generating an average force of 2.8 pN, which is much higher than that reported in most cases for single, randomly oriented HMM[11,41], except when measurements were made with a relatively high trap stiffness[9,42,43]. In isotonic contraction, $N_a$ decreases almost in proportion to $F$ (Fig. 4e, continuous line), in remarkable agreement with in situ experiments[8].

## Discussion

The nanomachine described in this paper sets the forefront of the research on the mechanics of a limited ensemble of muscle myosins. The performance of our system emerges clearly from the comparison with the ongoing research based on alternative approaches.

In theory, the best way to obtain an oriented ensemble of myosin motors consists in using either a native isolated thick filament or a synthetic cofilament made by rods and myosin molecules. With this motor design and the laser trap Kaya and collaborators obtained, in ATP-free solution, static stiffness measurements in agreement with fibre measurements[8,12,22]. However, the active performance of their motor system (in 1 mM ATP)[13] consisted only in transient displacements of the actin filament abruptly interrupted after variable extent, without any production of steady force and shortening. Other approaches that exploited the Three Bead Assay geometry (originally designed for single molecule mechanics[9]) by increasing the myosin density on the surface of the fixed bead to have multiple motor interactions[41,44] did not achieve any physiological machine performance, as the [ATP] was systematically kept at least ten times smaller than the millimolar concentration to increase the lifetime of the interactions.

The design of our machine, which exploits the dual laser beam technique, appears a decisive choice for the success of the assay. Beyond the large dynamic range, the DLOT geometry ensures that the array of interacting motors lies on a plane that is parallel to the BTA, in this way preserving their condition of "in parallel

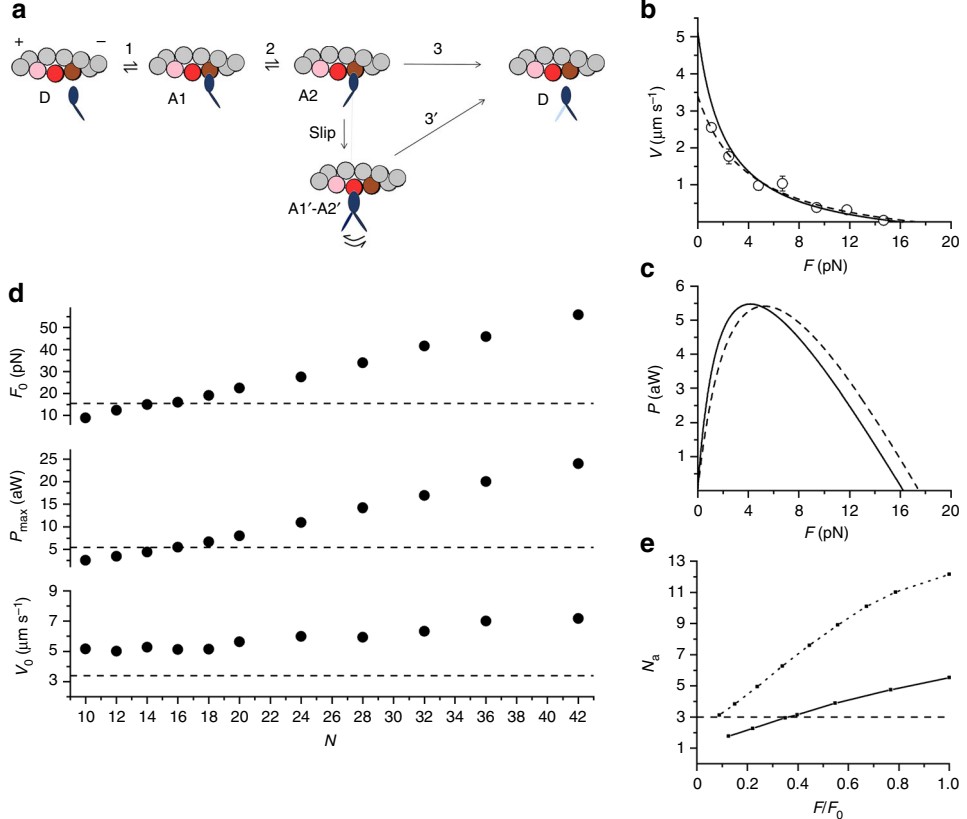

**Fig. 4** Model simulation of the mechanical output of the machine. **a** Kinetic scheme, with three states of the myosin head (blue): D, detached; A1 and A2, attached to one monomer of the actin filament (brown). The scheme includes the possibility that during shortening the attached head in the A2 state slips to the next actin monomer farther from the centre of the sarcomere (red). The probability of a second slipping to the pink monomer is limited to 1/10 of that of the first slipping. Details on the strain dependence of the rate functions and how they are constrained by fitting skeletal muscle performance are in Supplementary Fig. 6. The corresponding equations are listed in Supplementary Table 1. The effect of trap compliance on the nanomachine performance is detailed in Supplementary Fig. 7. **b** $F$–$V$ relation calculated for a number of available heads, $N = 16$ (continuous line) and experimental relation from Fig. 2e (open circles and dashed line). **c** Corresponding $P$–$F$ relations. **d** Dependence on $N$ of the three parameters featuring the machine performance, as indicated in the ordinate of each plot. The dashed lines are the respective experimental values. **e** Relation between number of attached heads ($N_a$) and force relative to $F_0$ calculated with the model for $N = 16$ (continuous line) and 32 (dotted line)

force generators". The lack of alignment between the actin filament and the motor ensemble in the single laser trap and in any other existing system based on single actin-single myosin filament[13–15] is likely the reason of their limited performance, as the attached head closest to the bead experiences an additional stress generated by the out-of-axis vertical component of the ensemble force. In ref. [13], the interactions between actin and myosin filaments produce transient displacements, lasting only ~100 ms, during which a force–velocity relation can be calculated using the time derivative of the displacement against the trap compliance, while force is continuously changing. However, a force–velocity relation calculated in this way is based on the not verified assumption that the number of available heads is constant during the whole interaction, and moreover, it does not fulfil the requirement that the force and the corresponding shortening velocity should be steady-state quantities produced by a collective motor based on muscle myosin.

Noteworthy, in contrast with the results from the two-filament design[12,13], but in agreement with in situ results[35], in our HMM-made machine low-load shortening does not imply any need of assuming a large reduction of stiffness in negatively strained attached heads. Eventually, a feature that makes our nano-machine design unique is that the actual length of the motor array is at least five times shorter than the length of the actin filament, so that the number of heads interacting with actin remains

constant for a total relative sliding distance of 2–3 μm, that is, one order of magnitude larger than in the native system in vivo, allowing the whole force–velocity relation to be determined during the same interaction.

The model simulation of the nanomachine performance predicts that, during shortening at the maximum power ($P_{max}$, 5.4 aW, attained at 0.3 $F_0$, Figs. 2f and 4c), the rate of ATP hydrolysis per myosin head ($\varphi_{Pmax}$) is ~30 s$^{-1}$ (Table 1). $N$ being 16 in our machine, the total ATP split per second by the machine at $P_{max}$ is (30 s$^{-1}$ × 16 =) 480 s$^{-1}$. The ratio between $P_{max}$ and the corresponding total ATP split per second is the mechanical energy delivered per ATP, that is (5400 zJ s$^{-1}$/480 s$^{-1}$ =) 11.2 zJ. Thus, with a free energy of the ATP hydrolysis of 110 zJ[24], the efficiency of the nanomachine ($\varepsilon$) would be (11.2/110 =) 0.1, at least three times lower than that reported for fast mammalian muscle (0.3–0.4)[24]. Taking into account that the random orientation of the motors reduces the force by 45%, the machine with correctly oriented motors would produce a work per ATP of 20 zJ, with $\varepsilon$ still significantly lower than that in the muscle. This can be explained considering that, with $N = 16$, during shortening against a load ≤0.3 $F_0$, $N_a$ decreases to <3 (Fig. 4e, continuous line), which results to be the minimum average number of attached heads that ensures a continuous interaction, as defined by the probability that at least one head exerts force on the actin filament at any time[6,45]. In fact, the model predicts that, although

**Table 1 Simulated mechanical and energetic parameters of the half-sarcomere and their modulation by the conditions imposed by the nanomachine**

| | $F_0$ (pN) | $a/F_0$ | $P_{max}$ (aW) | $\varphi_0$ (s⁻¹ per head) | $\varphi_{Pmax}$ (s⁻¹ per head) | $r_{s,0}$ (s⁻¹ per head) | $r_{s,Pmax}$ (s⁻¹ per head) |
|---|---|---|---|---|---|---|---|
| Compliance 0.01 nm pN⁻¹ | 433 ± 5 | 0.365 | 462 | 11.65 | 35.50 | 0.60 | 31.24 |
| Compliance 3.7 nm pN⁻¹ | 552 ± 1 | 0.234 | 437 | 14.50 | 33.82 | 15.60 | 23.87 |
| Compliance 3.7 nm pN⁻¹ + random | 312 ± 7 | 0.188 | 237 | 14.48 | 33.70 | 13.52 | 24.63 |

Upper row: in vivo series compliance; middle row: trap compliance; lower row: trap compliance with random orientation of motors
$F_0$ force per half-thick filament, $a/F_0$ relative value of the parameter $a$ of Hill's hyperbolic equation[21], $P_{max}$ maximum power, $\varphi$ flux through step 1 of the cycle in Fig. 4a, corresponding to the ATP hydrolysis rate either in isometric condition ($\varphi_0$) or at $P_{max}$ ($\varphi_{Pmax}$), $r_s$ slipping rate within the same ATPase cycle (step "slip" in Fig. 4a) in isometric condition ($r_{s,0}$) and at $P_{max}$ ($r_{s,Pmax}$)

$\varphi_{Pmax}$ is independent of $N$ (Fig. 5a), $\varepsilon$ increases with $N$ up to a value of 0.17 attained for $N > 32$ (Fig. 5b). In this case, the correction for the effect of the random orientation of motors would increase $\varepsilon$ to 0.31, similar to muscle values[24]. The reduction of $\varepsilon$ for $N < 32$ is accounted for by a corresponding reduction in the power delivered per head, ($P_{max}/N =$) $P_{max,h}$ (Fig. 5c) and is likely explained by considering that only for $N > 32$ even at low load $N_a$ remains >3 (Fig. 4e, dotted line), which is the condition for at least one attached head at any time[6,45].

The performance of our synthetic nanomachine, made of less than ten HMM molecules, and its interpretation with a kinetic model based on mammalian muscle performance[24–27] allow a complete recapitulation of muscle mechanics and energetics at the molecular level. The conditions for coordination between myosin motors that lead the half-sarcomere to an efficient power generation emerge explicitly: once the different working conditions of the nanomachine (series compliance and orientation of the motors in the ensemble) are taken into account, the efficiency and the power per head are identical to those in the muscle, provided that the number of heads available for the interaction with the actin filament ($N$) are ≥32, that is, assuming that the two heads of each dimer work independently, the number of HMM molecules is ≥16. As the motor array in our nanomachine spans 523 ± 175 nm, the structural condition for performance optimisation is one HMM available every ((523 ± 175)/15 =) 34.9 ± 11.7 nm of actin filament. This spatial frequency corresponds to the half-pitch of filamentous actin (36 nm), which, in a one-dimensional machine, is the periodicity at which actin monomers become accessible to myosin heads. Considering the limits of the present method of constructing the motor array, which provides an average spatial frequency of one HMM every ~70 nm (Fig. 1 and Supplementary Fig. 3), 32 available heads could be achieved only by doubling the span of the myosin array. How do the structural constraints for maximum power and efficiency found here for the one-dimensional machine apply to the three-dimensional architecture of the muscle sarcomere? The double-hexagonal lattice formed by myofilaments provides a 2:1 ratio between actin and myosin filaments, with the actin filament running in the centre of a triangular column formed by three neighbouring myosin filaments; along 700 nm of the myosin filament, there are (700 nm/14.3 nm × 3 =) 147 HMM dimers, that, at full overlap, yield (147/2 ~) 73 HMM dimers per actin filament with a spatial frequency of (73/700 nm ~) 0.1 HMM per nm. Thus the minimum length of the functional unit for maximum efficiency in situ is predicted to be (16/0.1 =) 160 nm, that is (160/523 =) 1/3 of that of the one-dimensional machine. This conclusion shows that, given the three-dimensional architecture of the sarcomere with respect to the one-dimensional machine, the length of the functional unit for collective motor action in the half-sarcomere at full filament overlap is (700/160 ~) 4 times greater than that necessary for a myosin head to work at maximum efficiency and power. A four times longer thick filament is

a treat for force as it provides a corresponding increase in the number of force-generating motors acting in parallel on the actin filament.

We conclude that the coordination between myosin motors that maximises power and efficiency in striated muscle is based on an architecture, shared by the half-sarcomere and our one-dimensional nanomachine, which simply requires that the target sites on the actin filament are exposed to at least 16 HMMs linked through their rod to a common mechanical ground without any specific geometry such as that of the myosin molecules on the thick filament. It is the actin filament that makes the functional selection, which, in the case of a one-dimensional machine, consists in one binding site every 36 nm. Notably, a similar conclusion was drawn for the unloaded sliding velocity of actin filaments over engineered filaments of non-muscle myosin[46]. In muscle myosin-based machines, a specific property emerges from motor coordination: during shortening an attached head can rapidly regenerate the working stroke within the same ATPase cycle, by slipping to the next actin monomer farther from the minus end (Fig. 4a), while the actin filament slides along for the action of the other attached heads. This function is preserved in our nanomachine, which, starting from purified proteins, allows the mechanics and energetics of the collective myosin II motor to be studied in the absence of cytoskeleton and regulatory proteins, the effects of which can then be selectively tested with different degrees of reconstitution. Eventually, the machine provides a new quantitative assay for investigating muscle diseases related to mutations in sarcomeric proteins and testing small molecule effectors as potential therapeutic tools.

## Methods

**Animals and ethical approval**. Adult male New Zealand white rabbits (4–5 kg), provided by Envigo, were housed at CeSAL, University of Florence, under controlled temperature (20 ± 1 °C), humidity (55 ± 10%) and illumination (light on for 12 h daily, from 7 a.m. to 7 p.m.). Food and water were provided ad libitum. All animals were treated in accordance with both the Italian regulation on animal experimentation (authorisation no. 956/2015 PR) in compliance with Decreto Legislativo 26/2014 and the EU regulation (directive 2010/63). For the preparation of proteins, rabbits were euthanised by injection of an overdose of sodium pentobarbitone (150 mg kg⁻¹) in the marginal ear vein, in accordance with Italian regulation on animal experimentation (authorisation no. 956/2015-PR).

**Preparation of proteins**. Myosin was purified from rabbit psoas muscle according to protocols based on established methods[47]; HMM was prepared by chymotryptic digestion of myosin[48] and stored at –80 °C after quick freezing in liquid nitrogen. Protein concentrations were determined spectrophotometrically using the extinction coefficients $\epsilon_{280 nm}$ (ml mg⁻¹ cm⁻¹) = 0.56 and 0.65 for myosin and HMM, respectively. The absorption was corrected for light scattering using the apparent absorption at 320 nm. Actin was prepared from acetone-extracted rabbit muscle powder[49]. For stabilisation against depolymerisation and for fluorescence imaging, F-actin was labelled with a molar excess of tetramethylrhodamine-isothiocyanate-phalloidin (Sigma-Aldrich)[50]. The correct orientation of the actin filament relative to the myosin ensemble was achieved by using the BTA method established in Ishiwata laboratory[16] and improved by Attila Nagy (National Institutes of Health, NIH, Bethesda, MD). The method exploits the property of gelsolin to cap the barbed (+end) of the filament. Gelsolin (plasma, porcine, Hypermol, Germany)

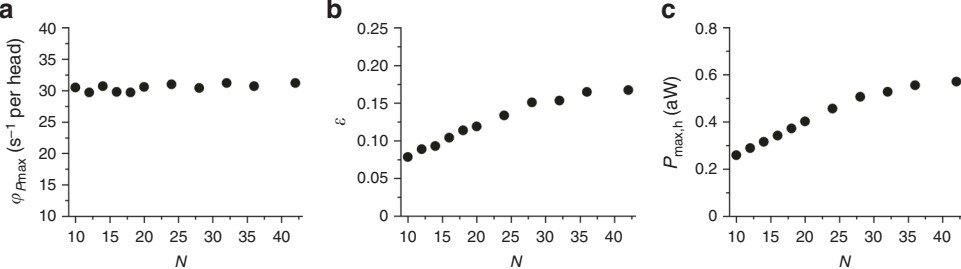

**Fig. 5** Model predictions of the energetic parameters. Dependence on $N$ of the various parameters is calculated at the maximum power ($P_{max}$). **a** Rate of ATP hydrolysis per myosin head, $\varphi_{Pmax}$. **b** Efficiency, $\varepsilon$. **c** Power per myosin head, $P_{max,h}$

was covalently bound to polystyrene beads with 1-ethyl-3-(3-dimethylaminopropyl)-carbodiimide in order to bind the +end of a single actin filament. Protein purity was evaluated with sodium dodecyl sulphate-polyacrylamide gel electrophoresis. The functionality of actin, myosin and HMM was tested with IVMA.

**Mechanical set-up**. The mechanics of the synthetic machine was measured using a DLOT apparatus[17] that allows measurements of force up to 200 pN (resolution 0.3 pN) and of movement up to 75,000 nm (resolution 1.1 nm). In the DLOT, two counter-propagating diode lasers (250 mW, 808 nm, Lumics GmbH., Germany) are focussed to form a single optical trap in the flow chamber[51]. The two laser beams have orthogonal polarisations, which allow their optical paths to be separated using polarising beam splitters, and are focussed through two water-immersion objective lenses (Olympus, UPLSAPO 60XW, NA 1.20) facing each other. The light exiting from either objective is projected on two position-sensitive detectors, which monitor the XY force components acting on a trapped bead (diameter 3 μm). The force measurement is based on the conservation of light momentum flux, so that it is unaffected by bead size, shape, refractive index and location[52]. The actin filament is attached to the trapped bead acting as a force transducer and the support of the array of myosin motors is integral to the flow chamber (Supplementary Fig. 1), the position of which is controlled by a three-way piezoelectric nanopositioner (nano-PDQ375, Mad City Lab, Madison, WI, USA) acting as a motor-displacement transducer. With our National Instruments board controller, the nanopositioner resolution is <1 nm. Bright-field images of the bead and the support, illuminated with a blue LED (470 nm), are detected using a 3/4-inch charge-coupled device (CCD) camera (Bitron VIDEO). The image of the fluorescent actin filament is recorded using a custom-built optical path for fluorescence excitation and detection (100 mW, 532 nm laser attenuated to 10 mM, Melles Griot MGL; 542 nm bandpass filter, Semrock FF01-542/20-25; ND filters to reduce sample power to ~5 mW; 5 MHz pixel-readout rate EEV CCD camera, Princeton Instrument Inc., Pentamax system, sensitive area 7.7 × 7.7 mm², 512 × 512 pixel²). The magnification of the fluorescence image is ~×300, corresponding to 80 nm per pixel. Movies are recorded at 4 frames s⁻¹ and stored in the computer for further analysis. Two National Instruments boards are used for instrument control and data acquisition with custom-written LabVIEW algorithms: a PCI 6030E board with an upper frequency limit of 100 kS s⁻¹ for signal generation and data acquisition and a PCI 6703 board to align the lasers and to modulate the flow into the cell by controlling solenoid valves. The force-displacement transducer system can be servo-controlled either in position or force feedback (Supplementary Fig. 2)[18]. The frequency response of the system in position feedback is limited by the rise time of the piezo-stage movement ($t_r$, measured from 10% to 90% of the step). $t_r$ has been minimised to ~2 ms. The frequency response in force feedback is limited also by the damping exerted on the bead by the viscosity of the medium. With a bead radius of 1.5 μm and a viscosity coefficient of 7 mPa s (for a solution with 0.5% w/v methylcellulose 400 cP) the bead responds to a stepwise change in force delivered as command signal with a displacement with $t_r$ ~ 20 ms. The piezo-stage is mounted on a three-way micrometre actuator for centimetre range movement.

**Protein assembly in the nanomachine**. The experimental flow chamber is 180 μm deep and is made of two coverglass slides separated by two layers of parafilm[51]. In the present version of the flow chamber (Supplementary Fig. 1), the parafilm was cut in order to obtain two separate compartments in which actin and myosin were flowed separately, thereby preventing uncontrolled protein interactions: the upper compartment (blue) was used to introduce the BTA; and the lower compartment (red), in which the support for the myosin motors is mounted, to introduce HMM. The support for the motors is the lateral surface of a single-mode optical fibre (780HP, Thorlabs Inc.), chemically etched in solution of HF (48% for 42–45 min at room temperature) to a diameter of ~4 μm and functionalised for HMM attachment by coating with nitrocellulose (1% w/v)[48]. The selection of this support was based on two criteria: the smoothness of the surface and the absence of formation of aggregates following HMM deposition, as tested by imaging with an Atomic Force Microscope (Cypher-S Atomic Force Microscopy, Asylum Research, Oxford Instruments) (Supplementary Fig. 8), and the ability to provide an efficient IVMA (see Supplementary Movie 1). In this respect, the flat tip of the single-mode optical

fibre, which following the etching acquired the desired dimension, had to be discarded owing to both the roughness of the surface (Supplementary Fig. 8) and the absence of any evidence of actin sliding in IVMA. Other kinds of optical fibres also failed to fulfil both criteria, very likely depending on the different composition of the glass that produced, after etching, a deterioration in the smoothness of the surface, compromising the possibility to have a regular HMM distribution. A planar surface with the desired dimension could not be achieved by using rods of squared section, because the etching introduced concavity of the surface.

The functionalised optical fibre was positioned in the lower compartment just before the confluence of the upper compartment, and the chamber was sealed, by heating the two parafilm layers, to prevent leakage (Supplementary Fig. 1). The lower compartment was filled by means of a syringe pump (flow velocity 400 μl min⁻¹) with 100 μg ml⁻¹ HMM and 1 μM G-actin dissolved in buffer A (25 mM imidazole pH 7.4, 33 mM KCl, 0.1 mM CaCl₂, 5 mM MgCl₂, 10 mM DTT and 2 mM ATP). Actin was introduced in order to neutralise the dead (rigor-like) HMM. After 5 min, the time required for the deposition of the HMM molecules on the fibre surface, the unbound HMM were washed out with ATP-free buffer A plus 0.5 mg ml⁻¹ bovine serum albumin (BSA). BTAs, dissolved in buffer A, were introduced using the upper compartment and trapped, with the optical tweezers, one by one at the intersection of the flows. A BTA with a single actin filament at least 6–7-μm long was selected. Then the flow chamber was moved across the x–y planes by means of the micro-positioner to bring the BTA close to the optical fibre. The final, precise positioning of the support with the motor array relative to the BTA was achieved under nanopositioner control. To start the experiment, buffer B (buffer A plus 0.1 mg ml⁻¹ glucose oxidase, 20 μg ml⁻¹ catalase, 5 mg ml⁻¹ glucose plus 0.5% w/v methylcellulose, 400 cP) was flowed at 10 μl min⁻¹. The presence of methylcellulose inhibited the lateral diffusion of F-actin[6], thereby minimising the probability that, in 2 mM ATP, the interaction terminates during low-force isotonic contractions or during the initial phases of force redevelopment following large releases. Methylcellulose (0.5% w/v) did not affect the mechanical and kinetic properties of the machine, as demonstrated also in Supplementary Fig. 5b by comparing the simulation of force development against the trap compliance in a medium with viscosity 1 mPa·s (control) and 7 mPa·s (corresponding to 0.5% w/v methylcellulose). All experiments were conducted at room temperature (23 °C).

**Mechanical protocols**. *(1) Rupture events in ATP-free solution*: The flow cell was filled with buffer A containing different [HMM] (0–100 μg ml⁻¹); following HMM binding to the fibre surface, an ATP-free buffer A containing 0.5 mg ml⁻¹ BSA was flowed in (about 10 times the flow chamber volume) to remove all the unbound HMM and ATP. Subsequently, ATP-free buffer B was introduced. Then, after allowing the actin–myosin bonds to be formed, the motor array was pulled at a constant velocity of 25 nm s⁻¹ in position feedback in the direction orthogonal to the actin–myosin interface (z axis, Fig. 1a)[19]. When the actin filament became taut and the first rigor bond was under force, the trapped bead was displaced from the trap centre along the direction of the actin filament. The force component along the x axis was measured and the bead displacement along the x axis was calculated knowing the trap stiffness (0.27 pN nm⁻¹). The bead movement along the z axis was measured using LabVIEW custom-made image analysis program based on previously developed protocols[42]. When a rigor bond was broken, the bead returned towards the trap centre, and the x and z components of the stepwise displacement were calculated. The total length of the motor array was calculated by summing all the stepwise displacements from the first to the last-but-one (see Supplementary Fig. 3).

*(2) Isometric and isotonic contractions in 2 mM ATP*: In the experiments on the active nanomachine, solution B containing 2 mM ATP was allowed to continuously flow (~4–5 μl min⁻¹). No artefact was detectable on the force trace due to the flow. The BTA was brought towards the motor array by moving the nanopositioner along the z axis under position feedback, and following the formation of the actin myosin interface, the ensemble of motors started to develop force under position clamp. After the maximum steady force was developed, the control was either switched to force clamp for imposing different loads or kept in position clamp to deliver rapid releases or stretches (Fig. 2c).

**Data analysis**. The velocity of shortening ($V$, μm s$^{-1}$) in the responses to reduction of force to $F$ values (pN) below the isometric value ($F_0$) was measured by the slope of the length trace (interpolated with a black line for clarity in Fig. 2b, c). The $F$–$V$ data were fitted with the hyperbolic Hill equation[21]: $(F + a)(V + b) = (V_0 + b)a$, where $a$ and $b$ are the distances of the asymptotes from the ordinate and abscissa, respectively, and $V_0$ (the ordinate intercept) estimates the maximum or unloaded shortening velocity. $a$ is a parameter that is used to express the degree of curvature of the relation (the greater the curvature the smaller the value of $a$). $a$ has the dimension of a force (pN) and, when normalised for $F_0$, is an index of the relative maximum power that can be delivered at intermediate forces[21]. The power output ($P$) at any force was calculated by the product between $F$ and $V$ and expressed as (μm s$^{-1}$)·pN $= 10^{-18}$ W = aW. Dedicated programs written in LabVIEW (National Instruments) and Origin 2015 (OriginLab Corporation) were used for the analysis. All data are expressed as mean ± SEM unless otherwise stated.

**Model simulation**. The results were simulated numerically by implementing a simplified version of the mechanical–kinetic model used to simulate transient and steady-state response of the muscle fibre[28,30]. The stochastic model estimates the probability distributions of potential results by allowing for random variation in inputs over time until the standard deviation of the result is <5%. The mechanical cycle of the motors is depicted in Fig. 4a. The state transitions as well as the strain of the attached motors are stochastically determined according to the kinetics defined below and in Supplementary Fig. 6 and in Supplementary Table 1. Each attached motor exerts, on the actin filament, a force that depends on its conformation and its relative position with respect to the actin monomer to which it is attached. The model operates either in position or force clamp. The iteration time in the calculation of the dependent variable, $\Delta t$ ($= 10^{-5}$–$10^{-6}$ s), depends on the stiffness of the system, which is mainly dictated by the trap compliance. Both the force generated by the motors, transmitted by the BTA, and the force of the optical trap act on the trapped bead. The motion of the bead was simulated with overdamped dynamics by using a drag coefficient that was calculated according to the bead radius and the viscosity of the medium. In this way, the bead displacement in response to a stepwise change in force showed a rise time $t_r$ similar to that measured experimentally (20 ms).

**Kinetics of state transitions**. The kinetic scheme used to simulate the output of the nanomachine is based on three states of the myosin motors (one detached, D, and two different force-generating attached states, A1 and A2, Fig. 4a and Supplementary Fig. 6a). The stiffness of the attached motors is 2 pN nm$^{-1}$, according to the most reliable in situ and in vitro data[22,23], and the kinetics of the state transitions are selected on the basis of the mechanical and energetic properties of the half-sarcomere as derived from studies of mammalian skeletal muscle at the same temperature[24–27].

Detached motors (D), with the hydrolysis products (ADP and inorganic phosphate, P$_i$) in the catalytic site, attach to the actin monomer (5 nm in diameter) (D→A1, step 1) at values of $d$ (the relative axial position between the motor and the actin monomer with $d = 0$ when the force in A1 is zero) ranging from −2.5 to 2.5 nm, according to the principle of the nearest-neighbour interaction. Following the state transition A1→A2 (step 2), the motors undergo the working stroke that accounts for the generation of force at $d = 0$ and its maintenance during shortening[28,29]. The working stroke, accompanied by Pi release, is a multistep reaction the extent and speed of which depend on the degree of shortening and external load[28,30]. During isotonic shortening, the speed of the working stroke is fast enough for the reaction to be considered at the equilibrium and the force of each A2 motor maintains the isometric value (5 pN) for a sliding distance of 5 nm and then reduces attaining 0 at 10 nm[28,29] (see Supplementary Fig. 6b). The A2 motors detach (A2→D, step 3) following ADP release and ATP binding with a speed that, under physiological [ATP] (≥2 mM), is dictated by the conformation-dependent kinetics of ADP release[7,8,30]. In order to maintain the scheme simple, the hydrolysis step and the repriming of the working stroke are incorporated in state D and contribute to the limited speed of the attachment reaction. The rate functions for the state transitions depend on $d$, the relative position between the motor and the actin monomer (with $d = 0$ when the motor in the A1 state develops zero force) as shown in Supplementary Fig. 6c–e and Supplementary Table 1. The rate functions are first defined to simulate the mechanics and energetics of the half-sarcomere as derived from muscle studies[24–27] and then used to fit the output of the nanomachine.

In isometric contraction, the rate-limiting step in the cycle is detachment from A2: under these conditions the rate of ATP splitting is minimum (~11 s$^{-1}$ per myosin head at room temperature[10,24,27]) and the fraction of motors attached (the duty ratio) is maximum (~0.3[22]). During steady shortening, the duty ratio decreases and the ATP splitting rate increases[24,27], due to the increase in the rate of motor detachment following the working stroke[2,7,8,28,30]. $a/F_0$, the estimate of the curvature of the $F$–$V$ relation, is ~0.36[25,26] (Supplementary Fig. 6f, black circles). The rate of ATP splitting increases no more than 3 times with the reduction of load[24], so that the curvature of the $F$–$V$ relation and the resulting maximum power can be predicted only by assuming that, during shortening, the attached myosin motors can rapidly regenerate the working stroke by slipping to the next actin farther from the centre of the sarcomere during the same ATPase cycle[10,28,30–33] (step "slip" in Fig. 4a) and undergoing A1'–A2' state equilibration according to step

2 kinetics[29,30]. Detachment from either A1' or A2' (step 3') implies ATP hydrolysis (Supplementary Fig. 6e).

The $F$–$V$ relation of a half-thick filament at full overlap (in which 294 motors can interact with the nearby actin filaments) calculated from the model simulation (Supplementary Fig. 6f, continuous line) perfectly fits that derived from published data on fast mammalian muscle[25,26] (filled circles), entailing a $P_{max}$ at 0.3 $F_0$ (Supplementary Fig. 6g, continuous line). The ATPase rate ($\varphi$), calculated by the flux through step 1, is 11.65 s$^{-1}$ per myosin head at $F_0$ ($\varphi_0$) and 35.5 s$^{-1}$ at $P_{max}$ ($\varphi_{P_{max}}$) that is three times $\varphi_0$, in agreement with the literature[24,27] (Table 1).

Applying the model simulation to the synthetic machine implies the introduction of a large compliance in series, which causes the strain-dependent kinetics of the attached motors to be governed by the push–pull experienced when actin slides away–toward the bead for the addition–subtraction of the force contribution by each motor (Supplementary Fig. 7, upper row, right panel). With the series compliance equal to 3.7 nm pN$^{-1}$, $F_0$ and $\varphi_0$ increase by ~25% and the slipping rate ($r_{s,0}$), which is near 0 with the small (muscle) series compliance, increases up to ~14 s$^{-1}$ per head (Table 1). During isotonic shortening, both $\varphi$ and $r_s$ increase and the effect of the large compliance on both parameters vanishes (Table 1), because in any case the reduction of motor strain increases the motor cycling kinetics. However, in the compliant system, the perturbation of the motor strain induced by the force contribution of each single motor is still present (Supplementary Fig. 7, lower row, right panel), causing only a minor reduction in both the curvature of the $F$–$V$ relation and $P_{max}$ (dashed lines in Supplementary Fig. 6f, g and Table 1).

## Data availability

The data that support the findings of this study and the computer code used to generate results that are reported in the manuscript are available within the paper and its supplementary information files or are available from the corresponding author upon reasonable request.

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

## Acknowledgements

We thank Gabriella Piazzesi and Massimo Reconditi for insightful comments on the manuscript; Marco Capitanio for bead displacement imaging analysis; Mario Dolfi and the staff of the mechanical workshop of the Department of Physics and Astronomy (University of Florence) for electronic and mechanical engineering support and James Sellers and Attila Nagy (NIH, Bethesda, USA) for initial support for the bead-tailed actin preparation. This work was supported by Istituto Italiano di Tecnologia, SEED-2009 (Italy), Ente Cassa di Risparmio di Firenze Project 2010.1402 and 2015.0902 (Italy), PRIN 2010/2011 Ministero dell'Istruzione, dell'Università e della Ricerca (Italy) and NVKP-16-1-2016-0017 National Heart Program (Hungary).

## Author contributions

V.L., P.B. and M.S.Z.K. conceived the study. I.P., L.M., V.L., D.C. and P.B. designed the experiment and implemented the procedure. I.P., P.B., L.M., L.S. and G.F. executed the experiments and analysed the data. L.B., V.L., P.B., G.B. and C.S. implemented the model simulation. D.C. and P.B. designed and fabricated the myosin supports and the sample chamber. M.S.Z.K. and T.B. performed atomic force microscopic analyses. V.L. wrote the paper with input from the other authors.

## Additional information

**Competing interests:** The authors declare no competing interests.

