## [Peer Review File · Nature Communications]

Reviewers' Comments:

Reviewer #1:

Remarks to the Author:

This manuscript has both experimental and theoretical components.

The experimental section describes the implementation of a loaded motility assay in which many myosin HMM interact with an actin filament presumably recapitulating sliding filaments in muscle. The authors refer to this as a "myosin II nanomachine". This experimental design has several advantages over existing assays. Namely, it limits myosin to a limited surface area on a fibre such that the number of myosin interacting with an actin filament is independent of the length of the filament. This is an important improvement because it holds the number of myosin heads interacting with an actin filament fixed during a given experiment. The authors demonstrate this to be the case through rupture experiments, but the distribution of numbers of myosin in Fig. 2d is relatively broad (contrary to the authors' claim) implying that there is considerable variability in myosin numbers from one experiment to the next.

The experimental geometry still falls short in truly recapitulating muscle shortening in that myosin HMM is used instead of myosin filaments, where myosin exhibit an asymmetric stiffness that more accurately mimics muscle shortening.

Overall, the net technical improvements over existing loaded motility assays are significant and advance the field. The experimental F-V curves are similar to those previously reported by the Warsaw lab only here the experiments were performed at physiological ATP. The experimental tension transient experiments are new for a loaded motility assay and advance the field.

While the quality of the experimental data is impressive, the experimental results are not unexpected. The major conclusions of the manuscript result from running the data through the filter of a theoretical framework and are thus several, model-dependent steps removed from their actual data. Because the model consists of many parameters and assumptions, the model-dependent conclusions are speculative and unconvincing..

Reviewer #2:

Remarks to the Author:

In this paper, a one-dimensional skeletal myosin motor ensemble was constructed by attaching an ensemble of HMM molecules onto a single-mode optical fiber with an actin filament-bead complex manipulated by Dual Laser Optical Tweezers. It demonstrated that ~10 myosin molecules are minimally needed for steady state force and shortening generations. This research group has explore the molecular mechanisms of muscle contractions at the forefront of muscle fiber study field for a few decades and now shows a new experimental approach to close the gap between single molecule and muscle fiber works. Thus, this paper has a potential to attract sufficient interest for publication, if several important issues are adequately addressed.

1. The new experimental design allows a myosin motor ensemble to generate steady-state force and shortening, which mimic isometric and isokinetic contractions of muscle fibers. In this sense, I am convinced that this experimental approach is unique to explore the molecular mechanism of muscle contraction in in-situ conditions. However, I do not understand why authors decided to use HMM rather than synthetic myosin filaments, of which myosin molecules are correctly oriented on one-half side so that the effect of random distribution on the force generation can be avoided in the model calculation. Authors argued that the DLOT geometry minimizes a stress caused by the vertical component of force ensuring its geometry as a parallel force generator. I agree with this, but still wonder why HMMs were chosen. If authors have a particular reason, it should be discussed as well.

2. There is no detail information on the kinetic model, such as equations for the state transition rate functions shown in Supplementary Fig. 6c-e. Is the working stroke size for A1'-A2' transition 5 nm? Is the actin binding site located at every 5.5 nm? Such details of model should be provided in the Supplementary Material. Also, I cannot understand the constant force at $d = -5$ to 2 nm in the force profile in Supplementary Fig. 6b. The force in myosin has to be changed as a function of motor strain, which highly depends on the dynamic interplay between force-generating myosin molecules and actin as described in the text (lines 198-200) and shown in Supplementary Fig. 7. The assumption must be justified.

3. In Fig. 4b, the experimental F-V relation are best fitted with $N=16$, which is suggested to be reasonable by considering both heads are operative and work independently at 2 mM ATP (L214-217). If I understand correctly from this argument, authors suggest that 8 myosin molecules on average can interact with an actin filament in the experimental condition and so, 16 myosin heads are practically available as an independent force generator at 2 mM ATP. Is this correct? However, the parameter, N , is defined as the number of motors in the ensemble (L211). I do not think that it is common to assume that two myosin heads interact independently with actin at high duty ratio. Meanwhile in discussion (L310-312), authors stated that... the number of motors available for the interaction with the actin filament are ≥ 32 , that is, the number of HMM molecules is ≥ 16 . This means that the number of HMMs arranged in this study is functionally equivalent to double of the same number of myosin molecules arranged in sarcomere. I am extremely confused here as well. For any reason, this argument is very critical and must be carefully justified, since all the results of the model were tested at $N = 16$ (Fig. 4). In the simple probabilistic model, the number of consecutive steps can be estimated from the following equation, $P(x, N) = [1 - (1-R)^{(N-1)}]^{(x-1)}$, where x and N are the number of consecutive steps and interacting molecules, and R and $P(x, N)$ are the duty ratio and the probability of observed steps, respectively. If $N=16$ and $R=0.1$ or 0.2 , the average number of steps is expected to be 4 or 20, respectively at $P=0.5$. In the optical trap experiment, the duty ratio is 0.05-0.1 at the beginning of acto-myosin interaction. Then, the duty ratio increases with increasing load. So, if the number of interacting molecule is ~ 16 , the number of consecutive step is expected to be more than 11 steps ($= 15 \text{ [pN]} \times 3.7 \text{ [nm/pN]} / 5 \text{ [nm]}$). However, if the number of the interacting molecule is 8 as estimated from the rapture experiment, the average number of steps is expected to be 2 or 4 at $R=0.1$ or 0.2 , respectively. Hence, it is an extremely low chance for 8 motors to perform 11 consecutive steps to reach the observed maximum force.

Specific and minor comments:

L16 and elsewhere: ... "synthetic" one-dimensional nanomechine ... implies myosin filament, but it is not. So, I would remove this word.

L72: Show the diameter of bead and briefly explain why DLOT is preferentially used rather than single trap laser beam.

Fig. 1: What is the criterion to define the rapture event? For example in Fig. 1b, three rapture events highlighted in the violet line are similar to the sharp drop in tension around $t=80$ s. However, this point was not picked. Describe the criterion to define the rapture point.

L133, 168: the abbreviation of in vitro motility assay, IVMA, was shown without annotation in the main text.

Reviewer #3:

Remarks to the Author:

In this manuscript, the authors perform an optical tweezers assay on skeletal muscle myosin II. A single bead-tailed actin filament is positioned near a glass needle support (an etched optical fiber). The needle is coated with multiple myosin II HMM molecules, and the authors argue that this geometry is similar to that in a muscle sarcomere, which also has multiple myosins binding each actin filament. Unlike other similar work, the needle support positions the myosins in line with the

actin filament. This geometry minimizes normal forces that would tend to pull the filament away from myosins. The authors see processive runs at saturating ATP, and generate force-velocity curves using feedback control of the stage position. By comparing the measured efficiency to that estimated from a simulation, the authors estimate that 32 myosins are accessible to actin in this geometry. The work is a nice technical effort, although it is oversold in places.

The authors argue that their needle geometry is essential for collecting continuous runs that are similar to the muscle sarcomere. However, ref. 13 shows processive runs up to 30 pN, and at saturating ATP. The comment that "...the active performance of their motor system (in 1 mM ATP) consisted only in transient displacements of the actin filament abruptly interrupted after variable extent, without any production of steady force and shortening" is not how I read their figure 1, which uses a fixed trap. Ruptures at high load are expected in these measurements, and indeed these ruptures also occur in force clamp systems when the stage reaches its programmed travel limit.

I also note that this manuscript uses methylcellulose. Could this crowding reagent be a more significant contributor to the long processive runs than the needle geometry? The use of methylcellulose and its caveats should be discussed in the manuscript.

The simulation is a tough section to digest, but could be improved through some reorganization. The purpose, as stated in the first sentence is far too vague: "To understand how the performance of the nanomachine recapitulates muscle contraction at the molecular level." Instead, lead off with the conclusion (from the end of the introduction, which the reader will have forgotten by now), that 32 motors sharing a common ground drive the filament. Then, you can explain up front how you generate a model with only one free parameter, N , the parameter of interest. Then you can discuss the other parameters, and how their values are informed / constrained through other data and experiments. This order will help the reader keep track of the important features of the model.

I would also like to see a further discussion of this experimental geometry vs. the double-hexagonal lattice found in muscle, and mentioned briefly at the end. It seems to me that this structural constraint may be even more important than the "in line" geometry. Would there be a way to test this idea experimentally using several needles? Speculation only here, please. No need to construct a new instrument, which would take far too much effort for this manuscript.

Minor comments:

I would call the instrument a counterpropagating dual beam tweezers. Some forms of dual beam tweezers use a separate detection laser, but both lasers are combined in a single objective. The counterpropagating tweezers has its own advantages and is more challenging to set up.

In the discussion: "The unequalled performance of our system emerges clearly...", "...appears a decisive choice for...". I would suggest that the authors be more conservative in their claims here.

Fig. 2 lists a Gaussian fit with too many significant figures.

Ref. 13 is missing a journal name.

Reply to the Editor and to reviewers (reviewers' questions in italic)

We thank the reviewers for their detailed reports, which gave us the opportunity for improvement of the paper and clarification of substantive points. There are questions in the reviewers' report that require a common response and we would like to answer to those questions in the following preliminary collective response to the Editor and the Reviewers. More specific questions are answered in detail in the responses to the Reviewers.

1. The first question concerns the definition of the assay. Its definition as a "synthetic machine mimicking the striated muscle" is etymologically correct under the conditions that the machine is made assembling the contractile muscle proteins in such a way that it is able to reproduce the performance of the striated muscle, namely steady force and shortening. In this respect we think that our assay fits the definition of a synthetic machine, in contrast to an assay that uses the myosin filament but does not reproduce muscle output.

2. Following the definition at point 1, the performance of the machine must be analysed in relation to that of the muscle sarcomere in vivo. For this we took the published mechanical and energetic data for fast mammalian skeletal muscle (containing the same myosin isoform as that in our assay) and adapted a simplified kinetic model to fit them. Given the evidence that skinned fibres from rabbit psoas do not reproduce the performance of intact muscle, the most reliable source of in vivo data was found to be the EDL muscle of the mouse at the same temperature (Ranatunga 1982,1984; Barclay 2015). First the relevant mechano-kinetic parameters of the model were selected to fit mechanics and energetics of the muscle in vivo, then the scheme was adapted to the conditions of the machine (trap compliance, random head orientation) and the output of the machine was fitted leaving as the only free parameter the number of motors N . Under these conditions we think that the criticism of reviewer 1 about the "presence of several model-dependent steps" and the recommendation of reviewer 3 to start the selection of model parameters from our experiment do not take into due account the premise at the start of this point. The assay must be tested against the in vivo sarcomere performance and, in this respect, the crucial question is whether the results can be fitted just scaling N by a given amount from the value in the thick filament in situ.

3. The power of this procedure is demonstrated by the following arguments: (1) the experimental results give a direct demonstration of the ability of the machine to produce steady force and shortening, but the relative power and efficiency result smaller than those in vivo; (2) the model shows that $N = 16$ (which coincides with the N predicted by the rupture events in rigor assuming that the two heads of a dimer work independently) is the value that best fits the observed results (Fig. 4d); (3) however, only with $N > 32$ the power and efficiency of muscle sarcomere are approached (Fig. 5). The reviewer 2 criticism (point 3, referring to L310-312) suggests that the question concerning the difference between N estimated in our assay (let's call it $N_{\text{obs}} = 16$) and N predicted by the model for the optimum power ($N_w = 2 \times N_{\text{obs}} = 32$) did not appear clear and we have further clarified this issue as detailed in the reply to the reviewer.

We believe that the conclusions drawn from the application of the model to the machine data are solid just because our simulation procedure implies that the model parameters are preliminarily constrained by published muscle mechanics and energetics. Only under this condition we can check if the synthetic machine recapitulates muscle contraction.

4. An important issue raised by the reviewers is the choice of using HMM rather than synthetic or native myosin filaments in which myosin molecules are correctly oriented. In this respect we want to stress the point that our design, which implies an extent of the myosin array ten times smaller than the length of the actin filament, is so far the only one able to provide a constant number of interacting motors for the whole experiment, allowing for the first time the F - V relation and the force transient following a large release to be determined during the same interaction. Any attempt to ensure these conditions using myosin filaments (either in our experiments or in the literature, see Discussion) were unsuccessful. Moreover, we think that there is a fundamental intrinsic limit in the myosin filament assay, versus our assay, which consists of the fact that in the myosin filament assay the degree of overlap with the actin filament and thus the number of interacting motors change continuously with sliding, preventing the possibility to reliably define all the steady and transient states of the motor array within the same experiment.

Reply to Reviewer #1

This manuscript has both experimental and theoretical components.

The experimental section describes the implementation of a loaded motility assay in which many myosin HMM interact with an actin filament presumably recapitulating sliding filaments in muscle. The authors refer to this as a "myosin II nanomachine". This experimental design has several advantages over existing assays. Namely, it limits myosin to a limited surface area on a fibre such that the number of myosin interacting with an actin filament is independent of the length of the filament. This is an important improvement because it holds the number of myosin heads interacting with an actin filament fixed during a given experiment.

The reason identified by the reviewer is the driving idea that so far has been pursued only with our design.

The authors demonstrate this to be the case through rupture experiments, but the distribution of numbers of myosin in Fig. 2d is relatively broad (contrary to the authors' claim) implying that there is considerable variability in myosin numbers from one experiment to the next.

Indeed, the variation in number of rigor ruptures in Fig. 1c accounts for the variation in the distribution of isometric forces and the underlying number of interacting motors in Fig. 2d, provided that each rupture in Fig. 1 is due to an HMM and the two myosin motors per HMM act independently in the active contraction. According to the Reviewer comment we have removed the definition of Gaussian distribution of forces as "relatively narrow".

The experimental geometry still falls short in truly recapitulating muscle shortening in that myosin HMM is used instead of myosin filaments, where myosin exhibit an asymmetric stiffness that more accurately mimics muscle shortening.

As explained in the preliminary collective response, our design, which implies an extent of the myosin array ten times smaller than the length of the actin filament, is so far the only one which fits the requirement to provide a constant number of interacting motors for the whole experiment, allowing the F - V relation and the force transient following a large release to be recorded during the same interaction.

Overall, the net technical improvements over existing loaded motility assays are significant and advance the field. The experimental F - V curves are similar to those previously reported by the Warshaw lab only here the experiments were performed at physiological ATP. The experimental tension transient experiments are new for a loaded motility assay and advance the field.

The limits of previous loaded motility assays have been discussed in a dedicated section in Discussion.

While the quality of the experimental data is impressive, the experimental results are not unexpected. The major conclusions of the manuscript result from running the data through the filter of a theoretical framework and are thus several, model-dependent steps removed from their actual data. Because the model consists of many parameters and assumptions, the model-dependent conclusions are speculative and unconvincing.

In relation to this critique we would like the reviewer to consider that, as made explicit in the title, in this prototypal system we were interested in selecting the most efficient procedure for the realisation of a synthetic machine able to mimic the striated muscle. The results are the demonstration of the success of the design. As explained in the preliminary response, we disagree with the opinion that the conclusions are separated from the data by several model-dependent steps. The model simulation uses only one free parameter to fit the results, since all the other parameters, apart from those related to the specific methodological conditions (trap compliance and random motor orientation), are constrained,

independently of the data, by the mechanics and energetics of muscle.

Reply to Reviewer #2:

In this paper, a one-dimensional skeletal myosin motor ensemble was constructed by attaching an ensemble of HMM molecules onto a single-mode optical fiber with an actin filament-bead complex manipulated by Dual Laser Optical Tweezers. It demonstrated that ~10 myosin molecules are minimally needed for steady state force and shortening generations. This research group has explored the molecular mechanisms of muscle contractions at the forefront of muscle fiber study field for a few decades and now shows a new experimental approach to close the gap between single molecule and muscle fiber works. Thus, this paper has a potential to attract sufficient interest for publication, if several important issues are adequately addressed.

1. The new experimental design allows a myosin motor ensemble to generate steady-state force and shortening, which mimic isometric and isokinetic contractions of muscle fibers. In this sense, I am convinced that this experimental approach is unique to explore the molecular mechanism of muscle contraction in in-situ conditions. However, I do not understand why authors decided to use HMM rather than synthetic myosin filaments, of which myosin molecules are correctly oriented on one-half side so that the effect of random distribution on the force generation can be avoided in the model calculation. Authors argued that the DLOT geometry minimizes a stress caused by the vertical component of force ensuring its geometry as a parallel force generator. I agree with this, but still wonder why HMMs were chosen. If authors have a particular reason, it should be discussed as well.

As explained in detail in the preliminary collective response to the Editor and Reviewers, our design, which implies an extent of the myosin array ten times smaller than the length of the actin filament, is so far the only one which provides a constant number of interacting motors for the whole experiment, allowing the F - V relation and the force transient following a large release to be determined during the same interaction. The random distribution of force generators is the cost we pay to fulfil the above condition. To control motor orientation is a next hard task to pursue in exploitation of the machine.

2. There is no detail information on the kinetic model, such as equations for the state transition rate functions shown in Supplementary Fig. 6c-e.

In fact, we now notice that there is not any explicit reference to Supplementary Table 1, in which are listed the equations expressing the d -dependence of the rate constants. Reference to the Table is now made in the text and in the legends of Fig. 4 and Supplementary Fig. 6.

Is the working stroke size for A1'-A2' transition 5 nm? Is the actin binding site located at every 5.5 nm? Such details of model should be provided in the Supplementary Material. Also, I cannot understand the constant force at $d=5$ to 2 nm in the force profile in Supplementary Fig. 6b. The force in myosin has to be changed as a function of motor strain, which highly depends on the dynamic interplay between force-generating myosin molecules and actin as described in the text (lines 198-200) and shown in Supplementary Fig. 7. The assumption must be justified.

The force profiles of the A2 state (Supplementary Fig. 6b) is due to the fact that in this state the motor undergoes the multistep working stroke that accounts for the fact that the force remains constant during sliding up to 5 nm, as demonstrated in situ by the T2 curve (Huxley and Simmons 1971; Lombardi and Piazzesi 1995). This is now made explicit in the text (first para in section "Explanation of the machine performance with a kinetic model ..." and in the Methods "Kinetics of state transition". The force profile of the A1 state is that of a purely elastic element; the rate of the multistep reaction in A2 keeps the different sub-state occupancy during isotonic shortening at equilibrium. This is why we can assume that the strain in the A2 motors remains constant up to 5 nm and then declines as the working stroke transition is exhausted. As noted by the Reviewer and described at L193-205 and in Supplementary Fig. 7 fluctuations in length

(either isometric or isotonic) are generated by the combination of force changes by state transition and the push pull due to the large trap compliance.

3. In Fig. 4b, the experimental F-V relation are best fitted with $N=16$, which is suggested to be reasonable by considering both heads are operative and work independently at 2 mM ATP (L214-217). If I understand correctly from this argument, authors suggest that 8 myosin molecules on average can interact with an actin filament in the experimental condition and so, 16 myosin heads are practically available as an independent force generator at 2 mM ATP. Is this correct?

Correct.

However, the parameter, N , is defined as the number of motors in the ensemble (L211). I do not think that it is common to assume that two myosin heads interact independently with actin at high duty ratio. Meanwhile in discussion (L310-312), authors stated that... the number of motors available for the interaction with the actin filament are ≥ 32 , that is, the number of HMM molecules is ≥ 16 . This means that the number of HMMs arranged in this study is functionally equivalent to double of the same number of myosin molecules arranged in sarcomere. I am extremely confused here as well. For any reason, this argument is very critical and must be carefully justified, since all the results of the model were tested at $N = 16$ (Fig. 4).

The question of the Reviewer is relevant, but we think is the result of a misunderstanding of the reasons behind the two values of N . As detailed at point 3 in the preliminary collective response to the Editor and the Reviewers the model simulation indicates that the best fit of data is obtained with $N = 16$, but also that the relative power and efficiency of the machine are lower than that of the sarcomere in situ and that the same performance is recovered increasing N to 32 (Fig. 5). The model shows also that the reduced performance with $N=16$ is due to the fact that the reduction of the duty ratio with the reduction of the load (and increase in shortening velocity) brings the number of attached motors (N_A) below 3 (Fig. 4e). In this case the condition of at least one motor attached at any time is no longer fulfilled and even if the methylcellulose preserves the continuity of interaction, the average sliding velocity and thus the power reduce (Uyeda et al, 1990; Harada et al, 1990).

In the simple probabilistic model, the number of consecutive steps can be estimated from the following equation, $P(x, N)=[1-(1-R)^{(N-1)}]^{(x-1)}$, where x and N are the number of consecutive steps and interacting molecules, and R and $P(x, N)$ are the duty ratio and the probability of observed steps, respectively. If $N=16$ and $R=0.1$ or 0.2 , the average number of steps is expected to be 4 or 20, respectively at $P=0.5$. In the optical trap experiment, the duty ratio is 0.05-0.1 at the beginning of acto-myosin interaction. Then, the duty ratio increases with increasing load. So, if the number of interacting molecule is ~ 16 , the number of consecutive step is expected to be more than 11 steps ($= 15 [pN] \times 3.7 [nm/pN] / 5 [nm]$). However, if the number of the interacting molecule is 8 as estimated from the rapture experiment, the average number of steps is expected to be 2 or 4 at $R=0.1$ or 0.2 , respectively. Hence, it is an extremely low chance for 8 motors to perform 11 consecutive steps to reach the observed maximum force.

Taking $N = 16$ (according to our conclusion) and not 8, the argument of the Reviewer sounds quite sensible. We found that in some experiments at the low initial force the interaction could terminate abruptly or show force fluctuations (as mentioned in the text, these records were discarded for the estimate of τ by exponential fitting of the force development). Similarly, during low load shortening ($<0.4 F_0$) in some cases the interaction terminated abruptly. These features were reproduced by the model with $N = 16$ or lower. Following the Reviewer comments we now have done some analysis on the simulated force development against the trap compliance when $N = 16$, concerning the probability of sudden failure and the number of interactions during the time ($= 3\tau$) necessary to attain $0.95 F_0$ (see Fig. R1). We found that, in the absence of failures (75% of cases) on average the number of interactions (number of steps according to the Reviewer) to attain $0.95 F_0$ is 75. Is it compatible with the prediction of reviewer equation? To answer this

question we need an estimate of the duty ratio during the force rise. With $\tau = 0.11$ s (legend of Fig. 3) and $F_0 = 16$ pN (Fig. 2e), the initial rate of force rise is 145 pN/s and the initial V is 0.538 $\mu\text{m/s}$. At this V , from the F - V relation (Fig. 2e) and the N_a - F/F_0 relation (Fig. 4e) it results an N_a of 3.7 and a duty ratio ($=3.7/16$) of 0.24. With $N = 16$ and duty ratio = 0.24 the Reviewer equation gives a number of consecutive steps of about 50. However, it must be considered that a duty ratio of 0.24 is the lower limit of the values experienced during the force development (as it is calculated on the initial highest rate of force rise). Taking the average duty ratio between the initial value and the F_0 value, we obtain $((0.24+0.32)/2=)$ 0.28 and the number of consecutive steps becomes 137, which is larger than the value of 75 obtained from the simulation of force development.

Specific and minor comments:

L16 and elsewhere: ... "synthetic" one-dimensional nanomechine ... implies myosin filament, but it is not. So, I would remove this word.

We would like to keep this definition for the reasons given in the preliminary collective response to Editor and Reviewers.

L72: Show the diameter of bead and briefly explain why DLOT is preferentially used rather than single trap laser beam.

The bead diameter is 3 μm as reported in Methods. As reported in the Discussion comparing ours and other loaded assays, the DLOT geometry allows precise alignment of machine elements avoiding that a vertical component of the collective force could act on the first actin-attached motor from the side of the bead. In We gave more details of the advantage of using the DLOT (force measurements unaffected by bead size, shape, refractive index and location) in Methods.

Fig. 1: What is the criterion to define the rupture event? For example in Fig. 1b, three rupture events highlighted in the violet line are similar to the sharp drop in tension around $t=80$ s. However, this point was not picked. Describe the criterion to define the rupture point.

In Fig. R2 the violet trace from Fig. 1b is reported at a faster time base to make more explicit the criterion for discarding force drops as that at $t \sim 82$ s, which do not have the characteristic of rupture events. The force drop must be complete in less than 50 ms to be taken as a rupture event and this criterion is now in the legend of Fig. 1.

L133, 168: the abbreviation of in vitro motility assay, IVMA, was shown without annotation in the main text.

Annotation made at L168.

Reply to Reviewer #3:

In this manuscript, the authors perform an optical tweezers assay on skeletal muscle myosin II. A single bead-tailed actin filament is positioned near a glass needle support (an etched optical fiber). The needle is coated with multiple myosin II HMM molecules, and the authors argue that this geometry is similar to that in a muscle sarcomere, which also has multiple myosins binding each actin filament. Unlike other similar work, the needle support positions the myosins in line with the actin filament. This geometry minimizes normal forces that would tend to pull the filament away from myosins. The authors see processive runs at saturating ATP, and generate force-velocity curves using feedback control of the stage position. By comparing the measured efficiency to that estimated from a simulation, the authors estimate that 32 myosins are accessible to actin in this geometry. The work is a nice technical effort, although it is oversold in

places.

Please note that the difference in efficiency derives from the comparison with that of the muscle, not with that of the simulation.

The authors argue that their needle geometry is essential for collecting continuous runs that are similar to the muscle sarcomere. However, ref. 13 shows processive runs up to 30 pN, and at saturating ATP. The comment that "...the active performance of their motor system (in 1 mM ATP) consisted only in transient displacements of the actin filament abruptly interrupted after variable extent, without any production of steady force and shortening" is not how I read their figure 1, which uses a fixed trap. Ruptures at high load are expected in these measurements, and indeed these ruptures also occur in force clamp systems when the stage reaches its programmed travel limit.

The Reviewer misses to consider that the isometric force generated by an ensemble of myosins is expected to be maintained continuously (as far as ATP is available) at a steady level that depends on the number of motors in the ensemble. For a relatively small ensemble of motors like ours the maximum isometric force is the most stable condition, with respect to lower forces, because the duty ratio is higher. Only with single processive motors the probability of detachment increases at the stall force.

I also note that this manuscript uses methylcellulose. Could this crowding reagent be a more significant contributor to the long processive runs than the needle geometry? The use of methylcellulose and its caveats should be discussed in the manuscript.

The use of methylcellulose in a one-dimensional machine (in which the actin filament is not constrained in the trigonal lattice) serves to minimise the probability to terminate the interaction during low force isotonic contractions or following a large release imposed to record force redevelopment. Model simulation of force development against the trap compliance with and without the viscous drag exerted by methylcellulose (400 cP, 0.5%, Fig. R1) shows that statistically there is the same probability of sudden force drop to zero (which in the experiment without methylcellulose could imply termination of the interactions), indicating that, as already reported, the methylcellulose does not affect the kinetics and performance of the assay.

The simulation is a tough section to digest, but could be improved through some reorganization. The purpose, as stated in the first sentence is far too vague: "To understand how the performance of the nanomachine recapitulates muscle contraction at the molecular level." Instead, lead off with the conclusion (from the end of the introduction, which the reader will have forgotten by now), that 32 motors sharing a common ground drive the filament. Then, you can explain up front how you generate a model with only one free parameter, N , the parameter of interest. Then you can discuss the other parameters, and how their values are informed / constrained through other data and experiments. This order will help the reader keep track of the important features of the model.

As detailed at point 2 in the preliminary collective response to Editor and Reviewers our strategy for the model simulation is to test the performance of the synthetic machine in relation to that of the fast skeletal muscle that contains the same myosin isoform. To do this the best way is to constrain the parameters of the kinetic model to the muscle performance and then define the minimum changes that must be done to fit the machine performance. In this way we find that we can fit data simply by scaling down N .

I would also like to see a further discussion of this experimental geometry vs. the double-hexagonal lattice found in muscle, and mentioned briefly at the end. It seems to me that this structural constraint may be even more important than the "in line" geometry. Would there be a way to test this idea experimentally using several needles? Speculation only here, please. No need to construct a new instrument, which would take far too much effort for this manuscript.

As suggested by the Reviewer, the results triggered the interest to integrate the conclusions into the native muscle sarcomere. We believe that we have exploited our data adequately, being able to predict both the conditions that allow a one-dimensional machine to produce the same relative power and efficiency of the muscle sarcomere and the minimum length of the thick filament in situ for maximum efficiency.

Minor comments:

I would call the instrument a counterpropagating dual beam tweezers. Some forms of dual beam tweezers use a separate detection laser, but both lasers are combined in a single objective. The counterpropagating tweezers has its own advantages and is more challenging to set up.

The definition of counterpropagating lasers is already present in the description of the DLOT in Methods.

In the discussion: "The unequalled performance of our system emerges clearly...", "...appears a decisive choice for...". I would suggest that the authors be more conservative in their claims here.

We would like to leave the emphasis on the first in vitro myosin-based machine able of a performance comparable to that of striated muscle.

Fig. 2 lists a Gaussian fit with too many significant figures.

Amended.

Ref. 13 is missing a journal name.

Reference corrected.

	Number of records	Number of steps to get to $0.95F_0$	Number of records with force drop to 0
viscosity coefficient = 1 mPa·s (no methylcellulose)	72	75.5 ± 2.7	14
viscosity coefficient = 7 mPa·s (0.5% methylcellulose 400 cP)	68	72.5 ± 5.1	20

Figure R1 | Simulated force development against the trap compliance, with a viscosity coefficient of the medium of 1 mPa·s (control, left column) and of 7 mPa·s (0.5% methylcellulose, right column). Traces occasionally show a transient drop of force to 0, marking loss of acto-myosin interaction. Number of

available motors (N) = 16. The table reports the statistics about the number of steps to get to $0.95F_0$ and the number of records with force drop to 0 either in the control solution (upper row) or in solution with the viscosity coefficient increased by methylcellulose (lower row).

Figure R2 | Expanded force from Fig. 1b, violet. The segment can be identified in the original trace using the abscissa. The force drop mentioned by the Reviewer, at about 82 s as indicated by the horizontal bar, is not considered a rupture event because it doesn't show the abrupt transition (< 50 ms) as that shown by the following two force drops indicated by the vertical bars.

Reviewers' Comments:

Reviewer #1:

None

Reviewer #2:

Remarks to the Author:

I acknowledge that the authors have addressed all the comments and the questions. However, some of the authors' responses still need to be clarified to make this article suitable for publication.

1. This study showed an innovative improvement of experimental technique to demonstrate the minimum numbers of myosin molecules to perform the isotonic and isometric contractions. Then, the simulation model was developed to estimate the minimum number of molecules required for the efficiency observed in muscle. As reviewer #1 pointed out, the conclusion is model-dependent and thus, the authors need to show more effort to evaluate the simulation model. In the current manuscript, the simulation outputs, force, power and velocity, are compared with those obtained from experiments, while the mechanochemical steps defined in the simulation model were not vigorously tested. Thus, readers feel that the conclusion is speculative. I realized that individual steps generated by myosin motors cannot be identified from experimental data, but the authors nicely addressed my question regarding to the probabilistic model to predict the number of steps required to reach the stall force. This argument definitely strengthens the reliability of the mechanochemical steps in the simulation model. I highly recommend discussing this point briefly in the main text (the second paragraph of discussion on pages 11-12).

2. The authors justified the reason why HMMs were preferentially used rather than synthetic myosin filaments. This justification should be described in the text to make the experimental approach more convincing (page 11 lines 278-281). Moreover, I am still not sure why HMMs are better than full-length myosins. As pointed out by reviewer #1, full-length myosins may exhibit the effect of asymmetric stiffness on shortening speed at lower loads. This aspect should be discussed as well.

3. In Supplementary Fig. 6b, the plot is very confusing without the definition of "d". Although the parameter, d, is defined in Methods, it is better to define in the figure caption as well.

4. In lines 182-184, it is better to describe that the assumption of steady force production in state A2 during shortening (~5nm) was made based on the T2 curve obtained from in situ fibers (Huxley and Simmons 1971, Lombardi and Piazzesi 1995). Otherwise, it is hard to understand this biased curve except for people in muscle fiber study.

5. The definition of N is still ambiguous for me. In simulation, N is defined as the number of motors, however N is treated as the number of motor heads in experiment to make the estimated value consistent with experimental results by assuming that two motor heads work independently at high ATP concentration. In my best knowledge, some research groups rather suggest cooperativity between two heads (Tyska et al., 1999), while other groups argue that one of heads primarily contribute to force generations. Hence, I am not sure why the authors confidently assumed that two heads work independently at high ATP concentrations. If strong evidence exists, the reference should be cited. In any case, the argument that N = 32 myosin molecules in simulation corresponds to 16 HMM molecules in experiment is so confusing. One parameter for two definitions should be avoided.

6. In Fig. 1b, the unbinding points are determined based on the force drop completed in less than 50 ms. To define the force drop precisely, the lower limit value of force drop must be defined (e.g., $F > 5$ pN within 50 ms). The authors should describe this point as well.

Reviewer #3:

Remarks to the Author:

>> The authors argue that their needle geometry is essential for collecting
>> continuous runs that are similar to the muscle sarcomere. However, ref. 13
>> shows processive runs up to 30 pN, and at saturating ATP...

> The Reviewer misses to consider that the isometric force generated by an
> ensemble of myosins is expected to be maintained continuously (as far as ATP
> is available) at a steady level that depends on the number of motors in the
> ensemble. For a relatively small ensemble of motors like ours the maximum
> isometric force is the most stable condition, with respect to lower forces,
> because the duty ratio is higher. Only with single processive motors the
> probability of detachment increases at the stall force.

If I understand correctly, the difference between the two experiments is not so much in "steady force and shortening", it is the duration of the stall (note that in their fig. 3j, they show they get very close to stall at 1 mM ATP). If that is correct, then change the statement to the following, which I believe both accurately describes ref. 13 and captures the advances here:

"The active performance of their motor system allows them to record filament velocities over a range of loads, down to 5-10% of the unloaded values in 1 mM ATP. However, their displacements are transient, lasting only ~100 ms until detachment. In contrast, with our system we record loaded shortening velocities and approach to stall over many seconds, which allows us to alter the applied force within a single shortening event as shown in fig. 2."

>> I also note that this manuscript uses methylcellulose. Could this crowding
>> reagent be a more significant contributor to the long processive runs than the
>> needle geometry? The use of methylcellulose and its caveats should be
>> discussed in the manuscript.

> The use of methylcellulose in a one-dimensional machine (in which the actin
> filament is not constrained in the trigonal lattice) serves to minimise the
> probability to terminate the interaction during low force isotonic
> contractions or following a large release imposed to record force
> redevelopment. Model simulation of force development against the trap
> compliance with and without the viscous drag exerted by methylcellulose (400
> cP, 0.5%, Fig. R1) shows that statistically there is the same probability of
> sudden force drop to zero (which in the experiment without methylcellulose
> could imply termination of the interactions), indicating that, as already
> reported, the methylcellulose does not affect the kinetics and performance of
> the assay.

Thanks, this is what I wanted to know. Please insert this statement in the manuscript, somewhere around line 115 (before "The nanomachine responds with actin filament sliding..."). I am requiring this statement for publication because it appears to be one of the key differences between this work and the work in Refs 13 or 41 (in addition to the experimental geometry differences noted by the authors).

Re: simulation. My suggested edits do not change *how* the simulation was done or the conclusions, only how it is presented in the manuscript. The current

version "buries the lede". It can be easily improved in the manner I described in the first round. But this is up to you.

Re: "counterpropagating dual beam tweezers". I understand that the experiment is explained in the methods, I meant to say that you should insert the word "counterpropagating" at the first use of "dual beam laser tweezers".

Re: "The unequaled performance of our system emerges clearly...", "...appears a decisive choice for...". I will leave these statements for the editors to decide.

Reply to Reviewers (their reports are copied in italic)

Reply to Reviewer #2

I acknowledge that the authors have addressed all the comments and the questions. However, some of the authors' responses still need to be clarified to make this article suitable for publication.

1. This study showed an innovative improvement of experimental technique to demonstrate the minimum numbers of myosin molecules to perform the isotonic and isometric contractions. Then, the simulation model was developed to estimate the minimum number of molecules required for the efficiency observed in muscle. As reviewer #1 pointed out, the conclusion is model-dependent and thus, the authors need to show more effort to evaluate the simulation model. In the current manuscript, the simulation outputs, force, power and velocity, are compared with those obtained from experiments, while the mechanochemical steps defined in the simulation model were not vigorously tested. Thus, readers feel that the conclusion is speculative.

Let's note that Reviewer #2 here rises an issue (*the mechanochemical steps of the model were not vigorously tested*) that was not in his previous report but in that of reviewer #1. In any case we think that answering that question is of crucial importance in relation to our claim that by integrating results and model simulation we are able to recapitulate muscle performance with our nanomachine made by a small number of myosin motors. In this respect, as already explained in the response to Reviewer #1, the performance of the machine must be analysed in relation to that of the muscle sarcomere in vivo. For this, first of all, we took the published mechanical and energetic data from fast mammalian skeletal muscle and adapted a simplified kinetic model to fit them. Then the same scheme was applied to the conditions of the synthetic machine (trap compliance, random head orientation), keeping unaltered all the mechanokinetic parameters selected for fitting the muscle performance and leaving as the only free parameter the number of motors (N) available for interaction with the actin filament: from 294 per thick filament in the muscle half-sarcomere (under the assumption that the two heads of a myosin molecule act independently at the duty ratios estimated in vivo, ≤ 0.35), to a value to be established by fitting the nanomachine performance. This procedure, which has been further clarified in the revised text (lines 216-226) selected a value of $N = 16$. We believe that the conclusions drawn from the application of the model to the machine data are solid just because the procedure implies that the model parameters are preliminarily constrained by published muscle mechanics and energetics: only under this condition we can check if and how the synthetic machine recapitulates muscle contraction. The same number of available motors (16) was independently obtained by measuring the number of ruptures in rigor (8), under the assumption (Ref. 36) that in rigor both heads of each HMM are attached and share the stress, while in the active machine as in vivo the two heads of the same dimer work independently. In this respect see also point 5.

I realized that individual steps generated by myosin motors cannot be identified from experimental data, but the authors nicely addressed my question regarding to the probabilistic model to predict the number of steps required to reach the stall force. This argument definitely strengthens the reliability of the mechanochemical steps in the simulation model. I highly recommend discussing this point briefly in the main text (the second paragraph of discussion on pages 11-12).

We appreciate the suggestion of the reviewer and we added the agreement between number of steps predicted by the model for the rise of force against the trap compliance and those allowed on the basis of the probabilistic argument based on $N = 16$ at lines 249-253 (and the detailed analysis in Supplementary Fig. 5b). Actually we found this as a further argument in support of the solidity of the model.

2. The authors justified the reason why HMMs were preferentially used rather than synthetic myosin filaments. This justification should be described in the text to make the experimental approach more convincing (page 11 lines 278-281). Moreover, I am still not sure why HMMs are better than full-length myosins. As pointed out by reviewer #1, full-length myosins may exhibit the effect of asymmetric stiffness on shortening speed at lower loads. This aspect should be discussed as well.

Within the limits of the present technology, the reasons why our system is superior to the two-filament system are detailed in the Discussion at lines 266-291. Following the Reviewer suggestion, we have expanded the comparison between our HMM assay and synthetic myosin filament, even in relation to the absence in our model simulation of any need for the asymmetric stiffness of the motors at low load (lines 284-286).

3. In Supplementary Fig. 6b, the plot is very confusing without the definition of “ d ”. Although the parameter, d , is defined in Methods, it is better to define in the figure caption as well.

“ d ” definition put where suggested.

4. In lines 182-184, it is better to describe that the assumption of steady force production in state A2 during shortening ($\sim 5\text{nm}$) was made based on the T2 curve obtained from *in situ* fibers (Huxley and Simmons 1971, Lombardi and Piazzesi 1995). Otherwise, it is hard to understand this biased curve except for people in muscle fiber study.

T2 curve now is explicitly mentioned.

5. The definition of N is still ambiguous for me. In simulation, N is defined as the number of motors, however N is treated as the number of motor heads in experiment to make the estimated value consistent with experimental results by assuming that two motor heads work independently at high ATP concentration. In my best knowledge, some research groups rather suggest cooperativity between two heads (Tyska et al., 1999), while other groups argue that one of heads primarily contribute to force generations. Hence, I am not sure why the authors confidently assumed that two heads work independently at high ATP concentrations. If strong evidence exists, the reference should be cited. In any case, the argument that $N = 32$ myosin molecules in simulation corresponds to 16 HMM molecules in experiment is so confusing. One parameter for two definitions should be avoided.

We cannot find in the text the sentence underlined above that the Reviewer attributes to us. In any case we have further revised the text to remove any possibility of misunderstanding. At lines 217-220 we define N as the number of motors available for interaction with actin, mentioning the reasons for assuming that the two heads of a dimer work independently in the active muscle and thus that in the half thick filament of the muscle sarcomere N corresponds to the number of heads (294, that is twice the number of molecules). At lines 222-225 we deduce that in our system 8 HMM molecules at 2 mM ATP are expected to give $N = 16$ and this number, by the way, matches the value of N obtained by fitting the model to the machine performances. We now report also (lines 225-226) that previous work on single molecules suggest cooperativity between the two heads (Ref. 37). We don't think there is any confusion left in the text about the argument that our model simulation fit the machine performance with $N = 16$ and predicts that, to get the maximum efficiency observed *in situ*, the machine should have $N = 32$ (lines 304-313).

6. In Fig. 1b, the unbinding points are determined based on the force drop completed in less than 50 ms. To

define the force drop precisely, the lower limit value of force drop must be defined (e.g., $F > 5$ pN within 50 ms). The authors should describe this point as well.

Lower limit of force added according to the suggestion.

Reply to Reviewer #3:

*>> The authors argue that their needle geometry is essential for collecting
>> continuous runs that are similar to the muscle sarcomere. However, ref. 13
>> shows processive runs up to 30 pN, and at saturating ATP...*

*> The Reviewer misses to consider that the isometric force generated by an
> ensemble of myosins is expected to be maintained continuously (as far as ATP
> is available) at a steady level that depends on the number of motors in the
> ensemble. For a relatively small ensemble of motors like ours the maximum
> isometric force is the most stable condition, with respect to lower forces,
> because the duty ratio is higher. Only with single processive motors the
> probability of detachment increases at the stall force.*

If I understand correctly, the difference between the two experiments is not so much in "steady force and shortening", it is the duration of the stall (note that in their fig. 3j, they show they get very close to stall at 1 mM ATP). If that is correct, then change the statement to the following, which I believe both accurately describes ref. 13 and captures the advances here:

"The active performance of their motor system allows them to record filament velocities over a range of loads, down to 5-10% of the unloaded values in 1 mM ATP. However, their displacements are transient, lasting only ~100 ms until detachment. In contrast, with our system we record loaded shortening velocities and approach to stall over many seconds, which allows us to alter the applied force within a single shortening event as shown in fig. 2."

We think that the qualitative difference between the responses of Ref. 13 assay and our assay must be taken in due account for a correct comparison. In Ref. 13 there are no records of steady-state isometric force nor of steady shortening at constant velocity against a constant load lower than the isometric force. Steady-state responses are expected from an ensemble of myosin motors that is able to reproduce the collective motor properties of the muscle half-sarcomere and are uniquely reproduced by our synthetic machine. From these responses we can build a force-velocity curve which is a steady-state property of active muscle that defines the mechanical performance (power) and can be used to define the corresponding energetics.

Instead, in Ref. 13 the force-velocity relation is calculated by the time derivative of bead displacement (as referred in the last line of left column at page 12 of Ref. 13), while also force is continuously changing, contradicting the definition of the force-velocity curve and making quite indirect and uncertain to derive any steady-state kinetic and energetic information.

Reviewer 3 claims that in Fig. 3j of Ref. 13 it is shown that *they get very close to stall at 1 mM ATP*. Beside the fact that in Fig. 3j of Ref. 13 there is not a record but a calculated force-velocity relation, we think that here and elsewhere the nomenclature of single molecule mechanics (stall force) is unsuitable to describe the steady-state isometric force of an ensemble of myosin motors, in which a steady rate of ATP splitting accounts for the attachment-detachment cycles responsible for the steady force. Moreover, the isometric contraction corresponds to the condition for the maximum duty ratio, thus is the most stable response, as far as there is ATP. Instead, in records of Ref. 13 this condition is only occasionally approached and is

followed by rapid detachment, which, as explained in our paper, is likely due to lack of alignment between actin filament and motor array.

>> *I also note that this manuscript uses methylcellulose. Could this crowding reagent be a more significant contributor to the long processive runs than the needle geometry? The use of methylcellulose and its caveats should be discussed in the manuscript.*

> *The use of methylcellulose in a one-dimensional machine (in which the actin filament is not constrained in the trigonal lattice) serves to minimise the probability to terminate the interaction during low force isotonic contractions or following a large release imposed to record force redevelopment. Model simulation of force development against the trap compliance with and without the viscous drag exerted by methylcellulose (400 cP, 0.5%, Fig. R1) shows that statistically there is the same probability of sudden force drop to zero (which in the experiment without methylcellulose could imply termination of the interactions), indicating that, as already reported, the methylcellulose does not affect the kinetics and performance of the assay.*

Thanks, this is what I wanted to know. Please insert this statement in the manuscript, somewhere around line 115 (before "The nanomachine responds with actin filament sliding..."). I am requiring this statement for publication because it appears to be one of the key differences between this work and the work in Refs 13 or 41 (in addition to the experimental geometry differences noted by the authors).

Following the concern of Reviewer #3 about the role of 0.5% w/v methylcellulose we have further integrated the Methods section (lines 93-98) and introduced in Supplementary Material a detailed demonstration of the absence of effects on the kinetics of force development (see Supplementary Fig. 5b).

*Re: simulation. My suggested edits do not change *how* the simulation was done or the conclusions, only how it is presented in the manuscript. The current version "buries the lede". It can be easily improved in the manner I described in the first round. But this is up to you.*

See the detailed answer to Reviewer 2.

Re: "counterpropagating dual beam tweezers". I understand that the experiment is explained in the methods, I meant to say that you should insert the word "counterpropagating" at the first use of "dual beam laser\ tweezers".

We have added counterpropagating at the first use of DLOT (line 55), even if "dual counterpropagating laser optical tweezers" sounds awkward.

Re: "The unequalled performance of our system emerges clearly...", "...appears a decisive choice for...". I will leave these statements for the editors to decide.

We believe we have demonstrated that the originality of the paper justifies those statements.

Reviewers' Comments:

Reviewer #2:

Remarks to the Author:

Now, I am satisfied by all the answers except for one condition, which is the definition of the parameter, N. Additional descriptions made by the authors help to clarify the definition, but I think that the term, motor, generally represent a two-headed molecule. That's why I intuitively think that N represents a molecule for the first time. I suggest the authors redefining N as the number of "heads" rather than motors.

Reviewer #3:

Remarks to the Author:

The authors need to take both reviewers requests for changes to the main text of the manuscript more seriously. They tend to argue their points in the responses to the reviewers, and in some cases, those arguments have merit. But the best parts of that content-- from the reviewers or from the authors-- need to be added to the body text of the paper. Our job here is to improve your manuscript. Help us to help you.

Item 1

>>Thanks, this is what I wanted to know. Please insert this statement in the manuscript, somewhere around line 115 (before "The nanomachine responds with actin filament sliding..."). I am requiring this statement for publication because it appears to be one of the key differences between this work and the work in Refs 13 or 41 (in addition to the experimental geometry differences noted by the authors).

>Following the concern of Reviewer #3 about the role of 0.5% w/v methylcellulose we have further integrated the Methods section (lines 93-98) and introduced in Supplementary Material a detailed demonstration of the absence of effects on the kinetics of force development (see Supplementary Fig. 5b).

The authors changed an existing statement in the methods about methylcellulose (I actually prefer the old statement there). Instead, I am requiring them to place this statement in the main text. Just to be absolutely clear, "requiring" means "not optional". The statement is an observation (i.e., the system only works with added methylcellulose, which the competing groups did not use in their systems). As such, it belongs in a prominent location, not in the methods.

Please change the sentence at line 108 to say "...with 2 mM ATP and 0.5% w/v methylcellulose...", and please insert the statement from the methods immediately after this sentence. Note, this is a better location than what was suggested last round (around line 115).

Item 2

Re: "The active performance of their motor system allows them to record filament velocities over a range of loads, down to 5-10% of the unloaded values in 1 mM ATP. However, their displacements are transient, lasting only ~100 ms until detachment. In contrast, with our system we record loaded shortening velocities

and approach to stall over many seconds, which allows us to alter the applied force within a single shortening event as shown in fig. 2."

The authors respond with a digression about nomenclature ("stall" vs. "isometric contraction", etc.) which is all fine, but misses the point. I find the current wording about ref. 13 misleading, and the above sentences (at least the first two) to be a more balanced description of the situation. If the authors want to expand on differences between force clamps vs. fixed trap experiments, that is fine. Reword as appropriate. But the fact remains, the authors in reference 13 did measure loaded filament velocities down to 5% of the initial value, which many will consider to be isometric within a reasonable detection threshold.

Item 3

>>Re: "counterpropagating dual beam tweezers". I understand that the experiment is explained in the methods, I meant to say that you should insert the word "counterpropagating" at the first use of "dual beam laser tweezers".

>We have added counterpropagating at the first use of DLOT (line 55), even if "dual counterpropagating laser optical tweezers" sounds awkward.

Correct, that does sound awkward, because the adjectives are in the wrong order. Change to "counterpropagating, dual laser optical tweezers". "Counterpropagating" modifies "dual laser..." rather than "dual" modifying "counterpropagating...".

Reply to reviewers

All reviewers requests, and in particular those requiring their introduction in the main text, have been satisfied, as detailed below in the responses point-by-point (Reviewers' comments in italic). The modifications and integrations have been identified in the text by highlighting them in yellow.

Reviewer #2

Now, I am satisfied by all the answers except for one condition, which is the definition of the parameter, N . Additional descriptions made by the authors help to clarify the definition, but I think that the term, motor, generally represent a two-headed molecule. That's why I intuitively think that N represents a molecule for the first time. I suggest the authors redefining N as the number of "heads" rather than motors.

We too, now, understand the reasons of the ambiguity claimed by the reviewer and took his advice, changing motors with heads wherever necessary. Accordingly, the sentence at L 185-188, where N is defined, has been rewritten.

Reviewer #3

The authors need to take both reviewers requests for changes to the main text of the manuscript more seriously. They tend to argue their points in the responses to the reviewers, and in some cases, those arguments have merit. But the best parts of that content-- from the reviewers or from the authors-- need to be added to the body text of the paper. Our job here is to improve your manuscript. Help us to help you.

Item 1

>>Thanks, this is what I wanted to know. Please insert this statement in the manuscript, somewhere around line 115 (before "The nanomachine responds with actin filament sliding..."). I am requiring this statement for publication because it appears to be one of the key differences between this work and the work in Refs 13 or 41 (in addition to the experimental geometry differences noted by the authors).

>Following the concern of Reviewer #3 about the role of 0.5% w/v methylcellulose we have further integrated the Methods section (lines 93-98) and introduced in Supplementary Material a detailed demonstration of the absence of effects on the kinetics of force development (see Supplementary Fig. 5b).

The authors changed an existing statement in the methods about methylcellulose (I actually prefer the old statement there). Instead, I am requiring them to place this statement in the main text. Just to be absolutely clear, "requiring" means "not optional". The statement is an observation (i.e., the system only works with added methylcellulose, which the competing groups did not use in their systems). As such, it belongs in a prominent location, not in the methods.

Please change the sentence at line 108 to say "...with 2 mM ATP and 0.5% w/v methylcellulose...", and

please insert the statement from the methods immediately after this sentence. Note, this is a better location than what was suggested last round (around line 115).

We understand and agree with the reasons of the reviewer to clarify this discriminating methodological point and in particular that it must be treated in the main text as a preliminary point in the description of Results. For this reason, at L 81-84, we added the relevant sentence:

"0.5% w/v methylcellulose (400 cP) was present in the solutions used for the experiment to inhibit the lateral diffusion of F-actin⁶, thereby minimising the probability that, in 2 mM ATP, the interaction terminates during mechanical protocols that minimize the number of actin-attached heads."

Item 2

Re: "The active performance of their motor system allows them to record filament velocities over a range of loads, down to 5-10% of the unloaded values in 1 mM ATP. However, their displacements are transient, lasting only ~100 ms until detachment. In contrast, with our system we record loaded shortening velocities and approach to stall over many seconds, which allows us to alter the applied force within a single shortening event as shown in fig. 2."

The authors respond with a digression about nomenclature ("stall" vs. "isometric contraction", etc.) which is all fine, but misses the point. I find the current wording about ref. 13 misleading, and the above sentences (at least the first two) to be a more balanced description of the situation. If the authors want to expand on differences between force clamps vs. fixed trap experiments, that is fine. Reword as appropriate. But the fact remains, the authors in reference 13 did measure loaded filament velocities down to 5% of the initial value, which many will consider to be isometric within a reasonable detection threshold.

We understand the request of the reviewer to mention that also Ref. 13 describes, among the results, a Force-Velocity relation. Therefore at L 238-245 we have reported a revised version of the two recommended sentences, which we think clarifies the point, taking into account the request of the reviewer.

Item 3

>>Re: "counterpropagating dual beam tweezers". I understand that the experiment is explained in the methods, I meant to say that you should insert the word "counterpropagating" at the first use of "dual beam laser tweezers".

>We have added counterpropagating at the first use of DLOT (line 55), even if "dual counterpropagating laser optical tweezers" sounds awkward.

Correct, that does sound awkward, because the adjectives are in the wrong order. Change to "counterpropagating, dual laser optical tweezers". "Counterpropagating" modifies "dual laser..." rather than "dual" modifying "counterpropagating..."

Done (L 57).